# Free Hunch: Denoiser Covariance Estimation for Diffusion Models Without Extra Costs

**Severi Rissanen**
Aalto University
severi.rissanen@aalto.fi

**Markus Heinonen**
Aalto University

**Arno Solin**
Aalto University

## Abstract

The covariance for clean data given a noisy observation is an important quantity in many training-free guided generation methods for diffusion models. Current methods require heavy test-time computation, altering the standard diffusion training process or denoiser architecture, or making heavy approximations. We propose a new framework that sidesteps these issues by using covariance information that is *available for free* from training data and the curvature of the generative trajectory, which is linked to the covariance through the second-order Tweedie's formula. We integrate these sources of information using *(i)* a novel method to transfer covariance estimates across noise levels and *(ii)* low-rank updates in a given noise level. We validate the method on linear inverse problems, where it outperforms recent baselines, especially with fewer diffusion steps.

## 1 Introduction

Diffusion models (Sohl-Dickstein et al., 2015; Ho et al., 2020; Song et al., 2021) have emerged as a robust class of generative models in machine learning, adept of producing high-quality samples across diverse domains. These models function by progressively denoising data through an iterative process, learning to reverse a predefined forward diffusion process that systematically adds noise. Conditional generation extends the capabilities of diffusion models by allowing them to generate samples based on specific input conditions or attributes. This conditioning enables more controlled and targeted generation, making diffusion models applicable to a wide range of tasks such as text-to-image synthesis or linear inverse problems such as deblurring, inpainting, or super-resolution.

A strand of recent research has concentrated on applying pretrained diffusion models to accommodate user-defined conditions, enhancing the flexibility and control of a single model to an arbitrary number of tasks. These methods guide the sampler towards regions whose denoisings $p(\boldsymbol{x}_0 \mid \boldsymbol{x}_t)$ are compatible with the condition or constraint, which requires efficient denoising mean $\mathbb{E}[\boldsymbol{x}_0 \mid \boldsymbol{x}_t]$ and covariance $\mathbb{C}\mathrm{ov}[\boldsymbol{x}_0 \mid \boldsymbol{x}_t]$ estimates (Ho et al., 2022; Song et al., 2023a;b; Boys et al., 2023; Peng et al., 2024). While estimating the mean is straightforward through the denoiser, accurately determining the covariance has proven more challenging. Consequently, efficient approaches have been proposed with heavy approximations (Chung et al., 2023; Song et al., 2023a).

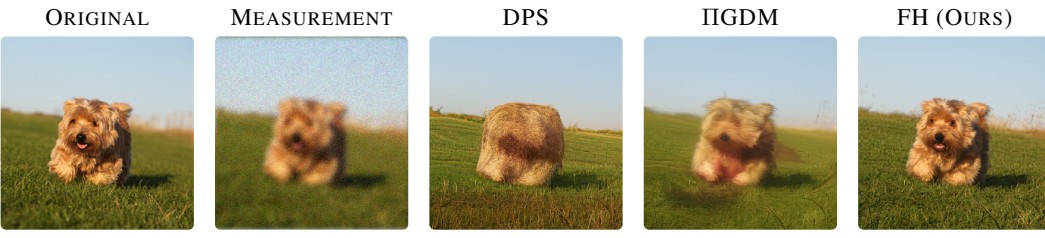

| Original | Measurement | DPS | ΠGDM | FH (Ours) |

Figure 1: Comparison of different conditional diffusion methods for deblurring, with a low number of solver steps (15 Heun iterations). DPS (Chung et al., 2023) and ΠGDM (Song et al., 2023a) work well with many steps, but accurate covariance estimates matter more for small step counts.

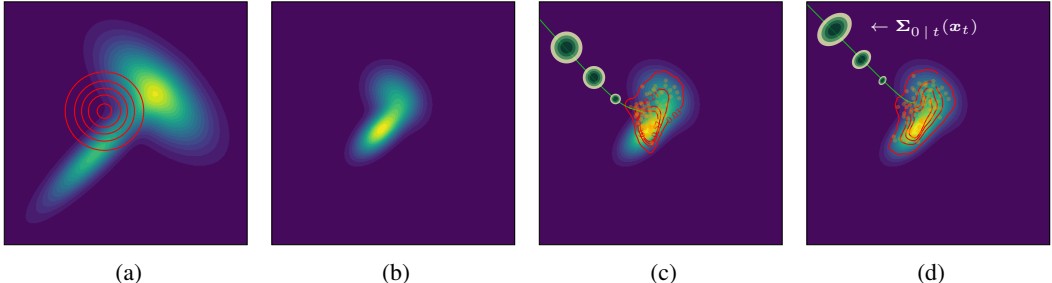

|     |     |     |     |
| --- | --- | --- | --- |
| (a) | (b) | (c) | (d) |

Figure 2: (a) A distribution $p(\boldsymbol{x}_0)$ represented by a pretrained diffusion model, and a Gaussian likelihood $p(\boldsymbol{y} \mid \boldsymbol{x}_0)$. (b) The (exact) posterior $p(\boldsymbol{x}_0 \mid \boldsymbol{y}) \sim p(\boldsymbol{x}_0)p(\boldsymbol{y} \mid \boldsymbol{x}_0)$. (c) Generated samples from a model with a heuristic diagonal denoiser covariance $\boldsymbol{\Sigma}_{0 \mid t}(\boldsymbol{x}_t)$, and a generative ODE trajectory with approximated $p(\boldsymbol{x}_0 \mid \boldsymbol{x}_t)$ shapes represented as ellipses along the trajectory. (d) Generated samples with our denoiser covariance.

In this paper, we propose a new method for denoiser covariance estimation, which we refer to as *Free Hunch* (FH). The name stems from the core insight that much of the required guiding covariance information is, in fact, freely available from the training data and the generative process itself. FH significantly improves accuracy over baselines, it is directly applicable to all standard diffusion models and does not require significant additional compute. This is achieved by integrating two sources of information into a unified framework: *(i)* the covariance of the data distribution and *(ii)* the implicit covariance information available in the denoiser evaluations along the generative trajectory itself. We apply the method to linear inverse problems, where we show mathematically that accurate covariance estimates are crucial for unbiased conditional generation, and achieve significant improvements over recent methods (see Fig. 1). In summary, our contributions are:

- **Methodological:** We propose a novel, efficient method for estimating denoiser covariances in diffusion models. It *(i)* does not require additional training, *(ii)* avoids the need for expensive score Jacobian computations, *(iii)* adapts to the specific input and noise level, and *(iv)* is applicable to all standard diffusion models.

- **Analytical:** We give a theoretical analysis of why accurate covariance estimation is crucial for reconstruction guidance in linear inverse problems.

- **Practical:** Our improved covariance estimates result in significant improvements over baselines in linear inverse problems, especially with small diffusion step counts.

## 2 BACKGROUND

Diffusion models are a powerful framework for generative modelling. Given a data distribution $p(\boldsymbol{x}_0)$, we consider the following sequence of marginal distributions:

$$p(\boldsymbol{x}_t) = \int \mathcal{N}(\boldsymbol{x}_t \mid \boldsymbol{x}_0, \sigma(t)^2 \boldsymbol{I}) p(\boldsymbol{x}_0) \, \mathrm{d}\boldsymbol{x}_0, \tag{1}$$

and corresponding reverse processes (Song et al., 2021; Karras et al., 2022)

$$\text{Reverse SDE:} \qquad \mathrm{d}\boldsymbol{x}_t = -2\dot{\sigma}(t)\sigma(t)\nabla_{\boldsymbol{x}_t} \log p(\boldsymbol{x}_t) \, \mathrm{d}t + \sqrt{2\dot{\sigma}(t)\sigma(t)} \, \mathrm{d}\omega_t, \tag{2}$$

$$\text{PF-ODE:} \qquad \mathrm{d}\boldsymbol{x}_t = -\dot{\sigma}(t)\sigma(t)\nabla_{\boldsymbol{x}_t} \log p(\boldsymbol{x}_t) \, \mathrm{d}t. \tag{3}$$

Here, the $\dot{\sigma}(t) = \frac{\mathrm{d}}{\mathrm{d}t}\sigma(t)$ and $\omega_t$ is a Brownian motion. The *score* $\nabla_{\boldsymbol{x}_t} \log p(\boldsymbol{x}_t)$ can be learned through score matching methods (Hyvärinen & Dayan, 2005; Vincent, 2011; Song et al., 2021). Starting at a sample $\boldsymbol{x}_t \sim \mathcal{N}(\boldsymbol{x}_t \mid \boldsymbol{x}_0, \sigma_{\max}^2 \boldsymbol{I})$ at a sufficiently high $\sigma_{\max}$ and integrating either differential equation backwards in time, we recover the data distribution $p(\boldsymbol{x}_0)$ if the score is accurate.

In conditional generation, we need to define the conditional score

$$\nabla_{\boldsymbol{x}_t} \log p(\boldsymbol{x}_t \mid \boldsymbol{y}) = \nabla_{\boldsymbol{x}_t} \log p(\boldsymbol{x}_t) + \nabla_{\boldsymbol{x}_t} \log p(\boldsymbol{y} \mid \boldsymbol{x}_t), \tag{4}$$

which decomposes into an unconditional score and the conditional adjustment through Bayes' rule. If we train a classifier to estimate the condition $\boldsymbol{y}$ given the noisy images $\boldsymbol{x}_t$, we get *classifier guidance* (Song et al., 2021; Dhariwal & Nichol, 2021). Using additional training compute for each conditioning task, however, may be prohibitive in some applications. A more modular way to do conditional generation is to define a constraint $p(\boldsymbol{y} \mid \boldsymbol{x}_0)$ only on the clean data points $\boldsymbol{x}_0$, and estimate

$$\nabla_{\boldsymbol{x}_t} \log p(\boldsymbol{y} \mid \boldsymbol{x}_t) = \nabla_{\boldsymbol{x}_t} \log \int \underbrace{p(\boldsymbol{y} \mid \boldsymbol{x}_0)}_{\text{constraint}} \underbrace{p(\boldsymbol{x}_0 \mid \boldsymbol{x}_t)}_{\text{denoise}} \, \mathrm{d}\boldsymbol{x}_0 = \nabla_{\boldsymbol{x}_t} \log \mathbb{E}_{p(\boldsymbol{x}_0 \mid \boldsymbol{x}_t)}\big[p(\boldsymbol{y} \mid \boldsymbol{x}_0)\big]. \quad (5)$$

That is, we need to integrate over all possible denoisings of $\boldsymbol{x}_t$ and their constraints. We want to avoid costly repeated sampling $\boldsymbol{x}_0 \mid \boldsymbol{x}_t$ from the reverse and seek practical approximations to $p(\boldsymbol{x}_0 \mid \boldsymbol{x}_t)$. A common approach is a Gaussian approximation

$$p(\boldsymbol{x}_0 \mid \boldsymbol{x}_t) \approx \mathcal{N}(\boldsymbol{x}_0 \mid \boldsymbol{\mu}, \boldsymbol{\Sigma}), \quad (6)$$

which is appealing since the so-called *Tweedie's formula* (Efron, 2011) links the score function $\nabla_{\boldsymbol{x}_t} \log p(\boldsymbol{x}_t)$ to exact moments of the posterior $p(\boldsymbol{x}_0 \mid \boldsymbol{x}_t) = \frac{p(\boldsymbol{x}_0)p(\boldsymbol{x}_t \mid \boldsymbol{x}_0)}{p(\boldsymbol{x}_t)}$,

$$\mathbb{E}[\boldsymbol{x}_0 \mid \boldsymbol{x}_t] = \boldsymbol{x}_t + \sigma_t^2 \nabla_{\boldsymbol{x}_t} \log p(\boldsymbol{x}_t), \quad (7)$$

$$\mathbb{C}\mathrm{ov}[\boldsymbol{x}_0 \mid \boldsymbol{x}_t] = \sigma_t^2 \Big(\sigma_t^2 \underbrace{\nabla_{\boldsymbol{x}_t}^2 \log p(\boldsymbol{x}_t)}_{\text{Hessian}} + \boldsymbol{I}\Big). \quad (8)$$

The correct mean (Eq. (7)) of the denoiser is directly implied by the score function, as long as our estimate of the score is accurate. Estimating the Tweedie covariance through the full Hessian in Eq. (8) is very expensive for high-dimensional data, however, and multiple methods have been proposed.

## 2.1 Related Work

**Denoiser covariance estimation in diffusion models** Previous attempts to improve denoiser covariance estimation in diffusion models can be broadly categorized into four categories:

1. **Heuristic methods:** Many methods (Ho et al., 2022; Song et al., 2023a;b) use a heuristic scaled identity covariance or can be seen as a special case where the covariance is zero (Chung et al., 2023). The methods are simple to implement but may result in biases in the conditional distributions.

2. **Training-based methods:** These involve training neural networks to directly output covariance estimates (Nichol & Dhariwal, 2021; Meng et al., 2021; Bao et al., 2022a; Peng et al., 2024). While potentially powerful, these approaches are not directly applicable to many existing diffusion models.

3. **Gradient-based methods:** These techniques estimate covariances by computing gradients of the denoiser (Finzi et al., 2023; Boys et al., 2023; Rozet et al., 2024; Manor & Michaeli, 2024b). However, getting the full covariance is computationally expensive and memory-intensive, making it challenging to apply to high-dimensional data without additional approximations. Manor & Michaeli (2024b) propose a method to compute the covariance along the top principal components efficiently.

4. **Post-hoc constant variance methods:** These approaches optimize constant variances for each time step based on pre-trained diffusion model scores (Bao et al., 2022b; Peng et al., 2024). While they do not require training or significant extra compute, they are limited in their ability to adapt to different inputs.

**Diffusion models for inverse problems and training-free conditional generation** Recent reviews can be found in Daras et al. (2024); Luo et al. (2024). Many works explicitly train conditional diffusion models for different tasks (Li et al., 2022; Saharia et al., 2022; Whang et al., 2022).

Many other methods adapt pre-trained diffusion models for inverse problems at inference time. DPS (Chung et al., 2022), ΠGDM (Song et al., 2023a), TMPD (Boys et al., 2023) and Peng et al. (2024); Song et al. (2023b); Ho et al. (2022) use backpropagation to explicitly approximate Eq. (5). We focus on these methods because they share a common framework with varying covariance approximations, enabling straightforward comparisons. Other methods modify the generative process to reduce the

residual $\boldsymbol{y} - \boldsymbol{A}\boldsymbol{x}_t$ in linear inverse problems (Song et al., 2021; Jalal et al., 2021; Choi et al., 2021). DDS (Chung et al., 2024) and DiffPIR (Zhu et al., 2023) determine guidance direction by optimizing for an $\boldsymbol{x}_0$ that balances proximity to both the measurement and denoiser output. DDNM (Wang et al., 2023) projects the denoised $\boldsymbol{x}_0$ to the null-space of the measurement operator during sampling. Peng et al. (2024) show that DDNM and DiffPIR can be framed in a similar framework. Rout et al. (2024) use second-order corrections to reconstruction guidance to mitigate biases in first-order Tweedie. Kawar et al. (2021; 2022) decompose the linear measurement operator with SVD to create specialized conditional samplers. Methods based on variational inference optimize for $\boldsymbol{x}_0$ that match with the observations while having high diffusion model likelihood (Mardani et al., 2024; Feng et al., 2023). Ben-Hamu et al. (2024); Wang et al. (2024b) optimize the noise latent $\boldsymbol{x}_T$ such that it matches with the observation. The methods by Wu et al. (2024); Dou & Song (2024); Trippe et al. (2023) frame conditional generation and inverse problems with a Bayesian filtering perspective, giving asymptotic guarantees with increasing compute.

**Other applications of Hessians and denoiser covariances**  Linhart et al. (2024) demonstrate that Gaussian approximation of $p(\boldsymbol{x}_0 \,|\, \boldsymbol{x}_t)$ enables compositional generation between diffusion models. Higher-order ODE solvers leverage the Hessian $\nabla_{\boldsymbol{x}_t}^2 \log p(\boldsymbol{x}_t)$ for efficient sampling (Dockhorn et al., 2022), while Sanchez et al. (2022) use it for causal discovery. Lu et al. (2022) improved model likelihoods by explicitly matching higher-order score gradients. Song & Lai (2024) identified the Hessian as equivalent to Fisher information for measuring step informativeness in conditional generation. Recent advances include efficient Hessian computation using training data (Wang et al., 2024a) and unsupervised audio editing via perturbing generation along denoiser covariance principal components (Manor & Michaeli, 2024a).

## 3 METHODS

We present our framework for incorporating prior data covariance information with curvature information observed during sampling. We define $\boldsymbol{\mu}_{0\,|\,t}(\boldsymbol{x}_t)$ and $\boldsymbol{\Sigma}_{0\,|\,t}(\boldsymbol{x}_t)$ as our approximations of $\mathbb{E}[\boldsymbol{x}_0 \,|\, \boldsymbol{x}_t]$ and $\mathbb{C}\mathrm{ov}[\boldsymbol{x}_0 \,|\, \boldsymbol{x}_t]$ at time $t$ and location $\boldsymbol{x}$. As we move from point $(\boldsymbol{x}, t)$ to $(\boldsymbol{x} + \Delta\boldsymbol{x}, t + \Delta t)$ in the diffusion process, the denoiser covariance changes but remains similar for small steps. We develop methods to transfer this information across time steps (Sec. 3.1), incorporate additional curvature information (Sec. 3.2), and combine these updates (Sec. 3.3). For high-dimensional data, we propose an efficient algorithm using diagonal and low-rank structures (Sec. 3.4). We discuss covariance initialization (Sec. 3.5) and in-

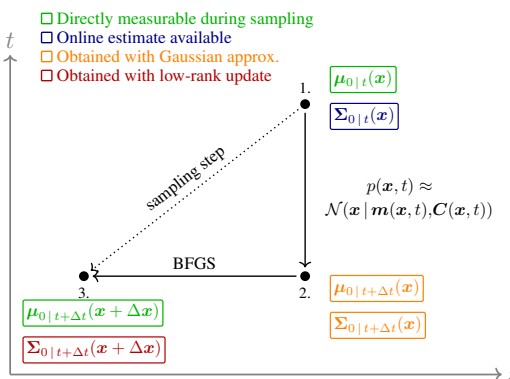

Figure 3: Sketch of our method during sampling.

troduce reconstruction guidance with a linear-Gaussian observation model (Sec. 3.6). Finally, we analyze why diagonal denoiser covariance overestimates guidance for correlated data at large diffusion times and demonstrate this issue with image data, showing that the problem is resolved with correct covariance estimation (Sec. 3.7).

**Notation**  In the following, we interchangeably use $p(\boldsymbol{x}, t)$ in place of $p(\boldsymbol{x}_t)$ where we want to emphasise the possibility to change either $\boldsymbol{x}$ or $t$, but not the other. However, in contexts where we talk about the posterior, we use $p(\boldsymbol{x}_t)$ and $p(\boldsymbol{x}_0 \,|\, \boldsymbol{x}_t)$ to emphasise the difference between the two random variables $\boldsymbol{x}_0$ and $\boldsymbol{x}_t$.

### 3.1 TIME UPDATE

Our goal is to obtain the evolution of the denoiser moments $\boldsymbol{\mu}_{0\,|\,t+\Delta t}(\boldsymbol{x}_t)$ and $\boldsymbol{\Sigma}_{0\,|\,t+\Delta t}(\boldsymbol{x}_t)$ (Eqs. (7) and (8)). The evolution of the moments under the diffusion process is characterised by the Fokker–Planck equation, and in practise intractable. We approximate the evolution with a second-order Taylor expansion of $\log p(\boldsymbol{x}_t)$ around point $\boldsymbol{x}_t$, which leads to a Gaussian form for $p(\boldsymbol{x}_t)$:

$$p(\boldsymbol{x}_t') \approx \mathcal{N}\big(\boldsymbol{x}_t' \,|\, \boldsymbol{m}(\boldsymbol{x}_t, t), \boldsymbol{C}(\boldsymbol{x}_t, t)\big), \tag{9}$$

where (temporarily dropping out the subscript from $\boldsymbol{x}_t$ for clarity)

$$\boldsymbol{m}(\boldsymbol{x}, t) = \boldsymbol{x} - \nabla_{\boldsymbol{x}}^2 \log p(\boldsymbol{x}, t)^{-1} \nabla_{\boldsymbol{x}} \log p(\boldsymbol{x}, t), \tag{10}$$

$$\boldsymbol{C}(\boldsymbol{x}, t) = -[\nabla_{\boldsymbol{x}}^2 \log p(\boldsymbol{x}, t)]^{-1}. \tag{11}$$

The evolution of the Gaussian (Eq. (9)) under the linear forward SDE (Eq. (1)) has a closed form (Särkkä & Solin, 2019). With the forward process induced by Eq. (1), this results in

$$\boldsymbol{m}(\boldsymbol{x}, t + \Delta t) = \boldsymbol{m}(\boldsymbol{x}, t), \tag{12}$$

$$\boldsymbol{C}(\boldsymbol{x}, t + \Delta t) = \boldsymbol{C}(\boldsymbol{x}, t) + \Delta\sigma^2 \boldsymbol{I}, \tag{13}$$

where $\Delta\sigma^2 = \sigma^2(t + \Delta t) - \sigma^2(t)$. That is, the forward keeps the mean intact while increasing the covariance. Using the above equations we can derive Tweedie moment updates:

$$\boldsymbol{\mu}_{0\,|\,t+\Delta t}(\boldsymbol{x}_t) = \boldsymbol{x}_t + \sigma(t + \Delta t)^2 \left( \sigma(t + \Delta t)^2 \boldsymbol{I} - \frac{\Delta\sigma^2}{\sigma(t)^2} \boldsymbol{\Sigma}_{0\,|\,t}(\boldsymbol{x}_t) \right)^{-1} \left( \boldsymbol{\mu}_{0\,|\,t}(\boldsymbol{x}_t) - \boldsymbol{x}_t \right), \tag{14}$$

$$\boldsymbol{\Sigma}_{0\,|\,t+\Delta t}(\boldsymbol{x}_t) = \left( \boldsymbol{\Sigma}_{0\,|\,t}(\boldsymbol{x}_t)^{-1} + \Delta\sigma^{-2} \boldsymbol{I} \right)^{-1}. \tag{15}$$

where $\Delta\sigma^{-2} = \sigma(t + \Delta t)^{-2} - \sigma(t)^{-2}$. The complete derivations can be found in App. A. As $\Delta t$ approaches zero, the Gaussian approximation becomes increasingly accurate. This is because the solution to the Fokker–Planck equation (which simplifies to the heat equation for variance-exploding diffusion) is a convolution with a small Gaussian $\mathcal{N}(\boldsymbol{x}\,|\,\boldsymbol{0}, \sigma(t)^2 \boldsymbol{I})$. The integral $\int p(\boldsymbol{x}_t) \mathcal{N}(\boldsymbol{x}_t - \boldsymbol{x}_s\,|\,\boldsymbol{0}, \sigma(t)^2 \boldsymbol{I}) \, \mathrm{d}\boldsymbol{x}_s$ is dominated by values near $\boldsymbol{x}_t$, as the Gaussian rapidly diminishes further away.

## 3.2 SPACE UPDATE FOR ADDING NEW LOW-RANK INFORMATION DURING SAMPLING

We take inspiration from quasi-Newton methods (*e.g.*, BFGS, see Luenberger et al., 1984) in optimization, where repeated gradient evaluations at different points are used for low-rank updates to the Hessian of the function to optimize. The diffusion sampling process is similar: we gather gradient evaluations $\nabla_{\boldsymbol{x}_t} \log p(\boldsymbol{x}_t)$ at different locations, and could use them to update the Hessian $\nabla_{\boldsymbol{x}_t}^2 \log p(\boldsymbol{x}_t)$. The Hessian is then connected to the denoiser covariance via Eq. (8). Here, we derive an even more convenient method to update $\boldsymbol{\Sigma}_{0\,|\,t}(\boldsymbol{x})$ directly.

To use update rules like BFGS, $\boldsymbol{\Sigma}_{0\,|\,t}(\boldsymbol{x})$ should be the Jacobian of some function. Thankfully, we notice that $\mathbb{Cov}[\boldsymbol{x}_0\,|\,\boldsymbol{x}_t]$ is proportional to the Jacobian of expectation $\mathbb{E}[\boldsymbol{x}_0\,|\,\boldsymbol{x}_t]$:

$$\mathbb{E}[\boldsymbol{x}_0\,|\,\boldsymbol{x}_t]\sigma(t)^2 = \left( \nabla_{\boldsymbol{x}_t} \log p(\boldsymbol{x}_t)\sigma(t)^2 + \boldsymbol{x} \right)\sigma(t)^2, \tag{Eq. (7)} \quad (16)$$

$$\nabla_{\boldsymbol{x}_t}\left( \mathbb{E}[\boldsymbol{x}_0\,|\,\boldsymbol{x}_t]\sigma(t)^2 \right) = \left( \nabla_{\boldsymbol{x}_t}^2 \log p(\boldsymbol{x}_t)\sigma(t)^2 + \boldsymbol{I} \right)\sigma(t)^2 = \mathbb{Cov}[\boldsymbol{x}_0\,|\,\boldsymbol{x}_t]. \tag{Eq. (8)} \quad (17)$$

We can then directly formulate the finite difference update condition equation for our estimate $\boldsymbol{\Sigma}_{0\,|\,t}(\boldsymbol{x})$:

$$\sigma(t)^2\left( \boldsymbol{\mu}_{0\,|\,t}(\boldsymbol{x} + \Delta\boldsymbol{x}) - \boldsymbol{\mu}_{0\,|\,t}(\boldsymbol{x}) \right) \approx [\boldsymbol{\Sigma}_{0\,|\,t}(\boldsymbol{x} + \Delta\boldsymbol{x})]\Delta\boldsymbol{x}. \tag{18}$$

This allows us to use a BFGS-like update procedure for the covariance and inverse covariance

$$\boldsymbol{\Sigma}_{0\,|\,t}(\boldsymbol{x} + \Delta\boldsymbol{x}) = \boldsymbol{\Sigma}_{0\,|\,t}(\boldsymbol{x}) - \frac{\boldsymbol{\Sigma}_{0\,|\,t}(\boldsymbol{x})\Delta\boldsymbol{x}\Delta\boldsymbol{x}^\top \boldsymbol{\Sigma}_{0\,|\,t}(\boldsymbol{x})}{\Delta\boldsymbol{x}^\top \boldsymbol{\Sigma}_{0\,|\,t}(\boldsymbol{x})\Delta\boldsymbol{x}} + \frac{\Delta\boldsymbol{e}\Delta\boldsymbol{e}^\top}{\Delta\boldsymbol{e}^\top \Delta\boldsymbol{x}}, \tag{19}$$

$$\boldsymbol{\Sigma}_{0\,|\,t}(\boldsymbol{x} + \Delta\boldsymbol{x})^{-1} = (\boldsymbol{I} - \gamma\Delta\boldsymbol{x}\Delta\boldsymbol{e}^\top)\boldsymbol{\Sigma}_{0\,|\,t}(\boldsymbol{x})^{-1}(\boldsymbol{I} - \gamma\Delta\boldsymbol{e}\Delta\boldsymbol{x}^\top) + \gamma\Delta\boldsymbol{x}\Delta\boldsymbol{x}^\top, \tag{20}$$

where

$$\Delta\boldsymbol{e} = \sigma(t)^2\left( \boldsymbol{\mu}_{0\,|\,t}(\boldsymbol{x} + \Delta\boldsymbol{x}) - \boldsymbol{\mu}_{0\,|\,t}(\boldsymbol{x}) \right) \quad \text{and} \quad \gamma = \frac{1}{\Delta\boldsymbol{e}^\top \Delta\boldsymbol{x}}. \tag{21}$$

While other update rules exist, BFGS has the advantage that it preserves the positive-definiteness of the covariance matrix. We provide further discussion in App. I.

| **Algorithm 1:** Time update | **Algorithm 2:** Space update |
|---|---|
| **Input:** $\boldsymbol{\Sigma}_{0\,|\,t}(\boldsymbol{x}), \sigma(t+\Delta t), \sigma(t), \boldsymbol{\mu}_{0\,|\,t}(\boldsymbol{x})$ | **Input:** $\boldsymbol{\Sigma}_{0\,|\,t}(\boldsymbol{x}_t), \boldsymbol{\mu}_{0\,|\,t}(\boldsymbol{x}+\Delta\boldsymbol{x}),$ |
| | $\boldsymbol{\mu}_{0\,|\,t}(\boldsymbol{x}), \sigma(t), \Delta\boldsymbol{x}$ |
| 1 $\Delta(\sigma^{-2}) = \sigma(t+\Delta t)^{-2} - \sigma(t)^{-2}$ | 1 $\Delta\boldsymbol{e} =$ Eq. (21) |
| $\boldsymbol{\Sigma}_{0\,|\,t+\Delta t}(\boldsymbol{x})^{-1} = \boldsymbol{\Sigma}_{0\,|\,t}(\boldsymbol{x})^{-1} + \Delta(\sigma^{-2})\boldsymbol{I}$ | 2 $\gamma =$ Eq. (21) |
| 2 $\boldsymbol{\Sigma}_{0\,|\,t+\Delta t}(\boldsymbol{x}) = (\boldsymbol{\Sigma}_{0\,|\,t+\Delta t}(\boldsymbol{x})^{-1})^{-1}$ | 3 $\boldsymbol{\Sigma}_{0\,|\,t}(\boldsymbol{x}+\Delta\boldsymbol{x}) =$ Eq. (19) |
| 3 $\boldsymbol{\mu}_{0\,|\,t+\Delta t}(\boldsymbol{x}) =$ Eq. (14) | 4 **return** $\boldsymbol{\Sigma}_{0\,|\,t}(\boldsymbol{x}+\Delta\boldsymbol{x})$ |
| 4 **return** $\boldsymbol{\Sigma}_{0\,|\,t+\Delta t}(\boldsymbol{x}), \boldsymbol{\mu}_{0\,|\,t+\Delta t}(\boldsymbol{x})$ | |

### 3.3 COMBINING THE UPDATES FOR PRACTICAL SAMPLERS

Note that the update step requires $\boldsymbol{\mu}_{0\,|\,t}(\boldsymbol{x})$ evaluated at two $\boldsymbol{x}$ locations but with the same $t$. Usually, we only have $\boldsymbol{\mu}_{0\,|\,t}(\boldsymbol{x})$ at *different* $t$ during sampling, however. The solution is that we can combine the time updates with the space updates in any diffusion model sampler as follows: Let's say we have two consecutive score evaluations $\nabla_{\boldsymbol{x}} \log p(\boldsymbol{x}, t)$ and $\nabla_{\boldsymbol{x}} \log p(\boldsymbol{x}+\Delta\boldsymbol{x}, t+\Delta t)$. We first update the denoiser mean and covariance with the time update to get estimates of $\boldsymbol{\Sigma}_{0\,|\,t+\Delta t}(\boldsymbol{x})$ and $\boldsymbol{\mu}_{0\,|\,t+\Delta t}(\boldsymbol{x})$. Then we can update $\boldsymbol{\Sigma}_{0\,|\,t+\Delta t}(\boldsymbol{x})$ with the BFGS update since we have $\boldsymbol{\mu}_{0\,|\,t+\Delta t}(\boldsymbol{x})$ from the time update and $\boldsymbol{\mu}_{0\,|\,t+\Delta t}(\boldsymbol{x}+\Delta\boldsymbol{x})$ from the second score function evaluation and Eq. (7). This is visualized in Fig. 3, and the algorithms for updating the covariance are given in Alg. 1 and Alg. 2.

### 3.4 PRACTICAL IMPLEMENTATION FOR HIGH-DIMENSIONAL DATA

While the method described so far works well for low-dimensional data, storing entire covariance matrices in memory is difficult for high-dimensional data. Luckily, this is not necessary since we only perform low-rank updates to the covariance matrix. In practice, we keep track of the following representation of the denoiser covariance:

$$\boldsymbol{\Sigma}_{0\,|\,t}(\boldsymbol{x}) = \boldsymbol{D} + \boldsymbol{U}\boldsymbol{U}^{\top} - \boldsymbol{V}\boldsymbol{V}^{\top}, \tag{22}$$

where $\boldsymbol{D}$ is diagonal and $\boldsymbol{U}, \boldsymbol{V}$ are low-rank $N \times k$ matrices. This structure comes from the two outer products in the the BFGS update (positive and negative). The vectors $\frac{\Delta\boldsymbol{e}}{\sqrt{\Delta\boldsymbol{e}^{\top}\Delta\boldsymbol{x}}}$ and $\frac{\boldsymbol{\Sigma}_{0\,|\,t}(\boldsymbol{x})\Delta\boldsymbol{x}}{\sqrt{\Delta\boldsymbol{x}^{\top}\boldsymbol{\Sigma}_{0\,|\,t}(\boldsymbol{x})\Delta\boldsymbol{x}}}$ become new columns in $\boldsymbol{U}$ and $\boldsymbol{V}$ respectively. In App. B, we show that inverting this matrix structure yields another matrix of the same form: $\boldsymbol{\Sigma}_{0\,|\,t}(\boldsymbol{x})^{-1} = \boldsymbol{D}' + \boldsymbol{U}'\boldsymbol{U}'^{\top} - \boldsymbol{V}'\boldsymbol{V}'^{\top}$. Using two applications of the Woodbury identity, this computation only requires inverting $k \times k$ matrices rather than $N \times N$ ones, enabling efficient calculation of both $\boldsymbol{\Sigma}_{0\,|\,t}(\boldsymbol{x}+\Delta\boldsymbol{x})^{-1}$ and the time update inverse.

### 3.5 INITIALISATION OF THE COVARIANCE

Having established methods for representing and updating denoiser covariances, we address initialization. While one might consider the limit $t \to \infty$ where $p(\boldsymbol{x}_t) \to \mathcal{N}(\boldsymbol{x}_t \,|\, \boldsymbol{0}, \sigma(t)^2\boldsymbol{I})$ and $\nabla^2_{\boldsymbol{x}_t} \log p(\boldsymbol{x}_t) \to -\frac{\boldsymbol{I}}{\sigma(t)^2}$, this is suboptimal: although the Hessian approaches identity at high $t$, the denoiser covariance approaches the data covariance. We estimate this from the data and initialise the covariance to it. For high-dimensional data, we approximate this covariance as diagonal in the DCT basis: $\boldsymbol{\Sigma}_t(\boldsymbol{x}_t) = \boldsymbol{\Gamma}_{\text{DCT}}\boldsymbol{D}\boldsymbol{\Gamma}_{\text{DCT}}^{\top}$. This is justified by natural images being approximately diagonal in frequency bases (Hyvärinen et al., 2009). While alternatives like PCA could be used, we found the DCT-based method sufficient. We provide additional discussion on the DCT basis in App. I.

### 3.6 GUIDANCE WITH A LINEAR-GAUSSIAN OBSERVATION MODEL

If the observation model $p(\boldsymbol{y} \,|\, \boldsymbol{x}_0)$ is linear-Gaussian, the reconstruction guidance becomes

$$\nabla_{\boldsymbol{x}_t} \log p(\boldsymbol{y} \,|\, \boldsymbol{x}_t) \approx \nabla_{\boldsymbol{x}_t} \log \int \mathcal{N}(\boldsymbol{y} \,|\, \boldsymbol{A}\boldsymbol{x}_0, \sigma_y^2\boldsymbol{I})\mathcal{N}(\boldsymbol{x}_0 \,|\, \boldsymbol{\mu}_{0\,|\,t}(\boldsymbol{x}_t), \boldsymbol{\Sigma}_{0\,|\,t}(\boldsymbol{x}_t)) \, \mathrm{d}\boldsymbol{x}_0$$

$$= (\boldsymbol{y} - \boldsymbol{A}\boldsymbol{\mu}_{0\,|\,t}(\boldsymbol{x}_t))^{\top}(\boldsymbol{A}\boldsymbol{\Sigma}_{0\,|\,t}(\boldsymbol{x}_t)\boldsymbol{A}^{\top} + \sigma_y^2\boldsymbol{I})^{-1}\boldsymbol{A}\nabla_{\boldsymbol{x}_t}\boldsymbol{\mu}_{0\,|\,t}(\boldsymbol{x}_t), \tag{23}$$

where $\boldsymbol{A}$ is the linear measurement operator (e.g., blurring). $\boldsymbol{\mu}_{0\,|\,t}(\boldsymbol{x}_t)$ is obtained using Tweedie's formula, and $\boldsymbol{\Sigma}_{0\,|\,t}(\boldsymbol{x}_t) = \boldsymbol{\Sigma}_{0\,|\,t}$ is assumed constant with respect to $\boldsymbol{x}_t$ when taking the derivative.

The linear-Gaussian setting is valuable as it both represents many real-world problems (deblurring, inpainting) and provides analytic insights into $\boldsymbol{\Sigma}_{0\,|\,t}$ choices. We will show that simplistic denoiser covariance approximations lead to severe overestimation of the guidance scale in Eq. (23).

### 3.7 Issues with Diagonal Denoiser Covariance

In this section, we will focus on DPS (Chung et al., 2023) and ΠGDM (Song et al., 2023a) for the linear inverse problem case. Both can be cast as using the same formula with $\boldsymbol{\Sigma}_{0\,|\,t} = r_t^2 \boldsymbol{I}$ and different post-processing steps on the resulting $\nabla_{\boldsymbol{x}_t} \log p(\boldsymbol{y}\,|\,\boldsymbol{x}_t)$ approximation:

1. In DPS, $r_t^2 = 0$. The resulting $\nabla_{\boldsymbol{x}_t} \log p(\boldsymbol{y}\,|\,\boldsymbol{x}_t)$ is scaled with $\frac{\xi \sigma_y^2}{\|\boldsymbol{y}-\boldsymbol{A}\boldsymbol{x}_0\|}$ ($\xi$ is a hyperparameter).

2. In ΠGDM, $r_t^2 = \frac{\sigma(t)^2}{1+\sigma(t)^2}$. The resulting $\nabla_{\boldsymbol{x}_t} \log p(\boldsymbol{y}\,|\,\boldsymbol{x}_t)$ is further scaled with $r_t^2$.

Importantly, in the case where the resulting $\mathbb{E}[\boldsymbol{x}_0\,|\,\boldsymbol{x}_t, \boldsymbol{y}] = \boldsymbol{x}_t + \sigma(t)^2 \nabla_{\boldsymbol{x}_t} \log p(\boldsymbol{x}_t\,|\,\boldsymbol{y})$ approximation is outside the data range $[-1, 1]$, the modified score in both DPS and ΠGDM is clipped to keep the denoiser mean within this range. Other choices include $r_t^2 = \sigma(t)^2$ (Ho et al., 2022).

**The mismatch between simplistic covariance and the denoiser Jacobian**  Notice that according to Eq. (8), $\nabla_{\boldsymbol{x}_t} \boldsymbol{\mu}_{0\,|\,t}(\boldsymbol{x}_t) \approx \frac{\mathbb{C}\text{ov}[\boldsymbol{x}_0\,|\,\boldsymbol{x}_t]}{\sigma(t)^2}$, where $\mathbb{C}\text{ov}[\boldsymbol{x}_0\,|\,\boldsymbol{x}_t]$ is the real denoiser covariance. For real data like images, this denoiser covariance is highly non-diagonal due to pixel correlations. This creates tension with the inverse $(\boldsymbol{A}\boldsymbol{\Sigma}_{0\,|\,t}\boldsymbol{A}^\top + \sigma_y^2 \boldsymbol{I})^{-1}$ in Eq. (23), which assumes diagonal $\boldsymbol{\Sigma}_{0\,|\,t}$.

**A toy model**  For the denoising task $\boldsymbol{A} = \boldsymbol{I}$, consider images with perfectly correlated pixels (same color), giving $\mathbb{C}\text{ov}[\boldsymbol{x}_0] = \boldsymbol{J}$ where $\boldsymbol{J}$ is all ones. As $t \to \infty$, $\mathbb{C}\text{ov}[\boldsymbol{x}_0\,|\,\boldsymbol{x}_t] \to \mathbb{C}\text{ov}[\boldsymbol{x}_0]$. Assume that the observation $\boldsymbol{y}$ and the denoiser mean $\boldsymbol{\mu}_{0\,|\,t}(\boldsymbol{x}_t)$ are similarly vectors of ones $\vec{\boldsymbol{1}}$ scaled by a constant, and thus $\boldsymbol{y} - \boldsymbol{\mu}_{0\,|\,t}(\boldsymbol{x}_t) = a\vec{\boldsymbol{1}}$. Note that $\nabla_{\boldsymbol{x}_t} \boldsymbol{\mu}_{0\,|\,t}(\boldsymbol{x}_t) = \frac{\boldsymbol{J}}{\sigma(t)^2}$. Now, the guidance terms with $r_t^2 = \frac{\sigma(t)^2}{1+\sigma(t)^2}$ read:

$$\nabla_{\boldsymbol{x}_t} \log p(\boldsymbol{y}\,|\,\boldsymbol{x}_t) = a\vec{\boldsymbol{1}}^\top \frac{1}{1+\sigma_y^2} \frac{\boldsymbol{J}}{\sigma(t)^2} = \frac{aN}{(1+\sigma_y^2)\sigma(t)^2} \vec{\boldsymbol{1}}^\top, \tag{24}$$

Here $N$ is the data dimensionality. Two key issues emerge: (1) the per-pixel guidance term scales with the total pixel count, and (2) for typical values ($a \approx 1$, $\sigma_y^2 \ll 1$), the guidance becomes implausibly large. For a 1000×1000 image, $\sigma(t)^2 \nabla_{\boldsymbol{x}_t} \log p(\boldsymbol{y}\,|\,\boldsymbol{x}_t) \approx N\vec{\boldsymbol{1}}$, yielding values around $10^6$ per pixel—far beyond the $[-1, 1]$ data range. This issue is even worse in DPS where $r_t^2 = 0$. For ΠGDM, clipping the denoiser mean to $[-1, 1]$ prevents trajectory blow-up but loses information and introduces biases. The scaling factors in DPS also reduce but do not eliminate the problem.

**Solution with the correct covariance**  In contrast, the same issue does not occur if we use the correct denoiser covariance in the formula:

$$\nabla_{\boldsymbol{x}_t} \log p(\boldsymbol{y}\,|\,\boldsymbol{x}_t) \approx (\boldsymbol{y}-\boldsymbol{\mu}_{0\,|\,t}(\boldsymbol{x}_t))^\top (\mathbb{C}\text{ov}[\boldsymbol{x}_0\,|\,\boldsymbol{x}_t]$$

$$+ \sigma_y^2 \boldsymbol{I})^{-1} \frac{\mathbb{C}\text{ov}[\boldsymbol{x}_0\,|\,\boldsymbol{x}_t]}{\sigma_t^2}. \quad (25)$$

Clearly, if $\sigma_y \to 0$, the covariances cancel out. Thus, the scale of the calculated guidance does not cause issues. In App. C, we repeat this analysis without assuming $\sigma_y = 0$.

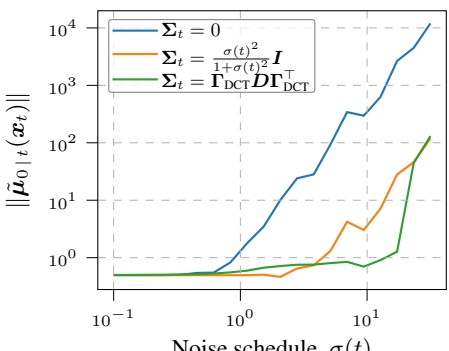

Figure 4: Norm of $\boldsymbol{\mu}_{0\,|\,t}(\boldsymbol{x})$ + $\sigma(t)^2 \nabla_{\boldsymbol{x}_t} \log p(\boldsymbol{y}\,|\,\boldsymbol{x}_t)$ for different co-variance estimation methods on ImageNet 256×256. Values >1 indicate overestimation since the data is normalized to $[-1, 1]$.

Fig. 4 showcases the issue in practice for a Gaussian blur operator $\boldsymbol{A}$ in ImagetNet 256×256 and a denoiser from Dhariwal & Nichol (2021). In comparison, the problem is less severe for a more sophisticated DCT-diagonal covariance approximation. However, even the DCT-diagonal method does cause the adjusted denoiser mean to diverge at high noise levels.



| (a) DPS | (b) ΠGDM | (c) ΠGDM (no $r_t^2$ scaling) | (d) FH (Ours) | (e) Optimal cov |

Figure 5: Different methods for posterior inference in the example in Fig. 2 and Jensen–Shannon divergences to the true posterior.

**Approximating** $\nabla_{\boldsymbol{x}_t}\boldsymbol{\mu}_{0\,|\,t}(\boldsymbol{x}_t)$  Given that the problem stems from the mismatch between $\nabla_{\boldsymbol{x}_t}\boldsymbol{\mu}_{0\,|\,t}(\boldsymbol{x}_t)$ and our covariance approximation, a solution needs to harmonize these two terms. Thus, we propose to further approximate $\nabla_{\boldsymbol{x}_0}\boldsymbol{\mu}_{0\,|\,t}(\boldsymbol{x}_t)$ by our denoiser covariance estimate when the scale of the guidance by calculating the full Jacobian is too large. In practice, we first calculate the adjusted denoiser covariance estimate $\boldsymbol{\mu}_{0\,|\,t}(\boldsymbol{x}_t)$, and fall back to approximating $\nabla_{\boldsymbol{x}_0}\boldsymbol{\mu}_{0\,|\,t}(\boldsymbol{x}_t) \approx \frac{\boldsymbol{\Sigma}_{0\,|\,t}(\boldsymbol{x}_t)}{\sigma(t)^2}$ in case our initial approximation $\|\sigma(t)^2 \nabla_{\boldsymbol{x}_t} \log p(\boldsymbol{y}\,|\,\boldsymbol{x}_t)\| > 1$ (which would push the trajectory in a direction that is outside the data range $[-1,1]$).

**Full algorithm.**  In App. D, we show full algorithms for linear inverse problems with our covariance estimation method, one with the Euler ODE solver and another that works with any solver.

## 4    EXPERIMENTS

We validate our method using synthetic Gaussian mixture model data and compare it against baselines on linear imaging inverse problems. Our experiments demonstrate that our more sophisticated covariance approximations reduce bias and improve results, particularly at lower diffusion step counts. We use a linear schedule $\sigma(t) = t$, as advocated by Karras et al. (2022), and follow their settings for our image diffusion models otherwise as well. For the image experiments, we used $\sigma_{\max} = 80$ and $\sigma_{\max} = 20$ for the synthetic data. We use a simple Euler sampler for the synthetic data experiments and a 2nd order Heun method (Karras et al., 2022) for the image experiments.

### 4.1    SYNTHETIC DATA EXPERIMENTS

**Toy data**  We first showcase the performance of different methods on a toy problem using a mixture of Gaussians distribution, which admits a closed-form formula for the score (see App. M). The results in Fig. 5 show that our method clearly outperforms DPS and ΠGDM, approaching the method using optimal covariance obtained by backpropagation and Eq. (8). Note that the example favours DPS, since we tuned the guidance hyperparameter for this particular task.

**The effect of dimensionality and correlation**  In Sec. 3.7, we noticed that the guidance scale is overestimated the larger the dimensionality is. A practical consequence is that the variance of the generative distribution can be underestimated. In App. E, we directly showcase this with synthetic data and show that it does not happen with our method.

**Approximation error in the covariance**  In App. G, we analyse the error in the covariance approximation for a low-dimensional example, and empirically show that the error approaches zero with a large amount of steps and a stochastic sampler.

### 4.2    IMAGE DATA AND LINEAR INVERSE PROBLEMS

We experiment on ImageNet 256×256 (Deng et al., 2009) with an unconditional denoiser from Dhariwal & Nichol (2021). We evaluate the models on four linear inverse problems: Gaussian deblurring, motion deblurring, random inpainting, and super-resolution. We evaluate our models with peak signal-to-noise ratio (PSNR), structural similarity index measure (SSIM, Wang et al., 2004) and learned perceptual image patch similarity (LPIPS, Zhang et al., 2018) on the ImageNet test set. We use the same set of 1000 randomly selected images for all models.

We solve the inverse in Eq. (23) using conjugate gradient, following Peng et al. (2024). Our custom PyTorch implementation uses GPU acceleration and adjusts solver tolerance based on noise levels. This optimization removes the solver bottleneck without noticeable performance loss. Details are in App. J.

**Proposed models** We introduce four new methods. The first, 'Identity', is initialized with identity covariance. 'Identity+Online' also uses the space updates. 'FH' is initialized with data covariance projected to a DCT-diagonal basis. Finally, 'FH+Online' enhances 'FH' with online updates.

**Baselines** We compare against several methods using Eq. (23) for linear imaging inverse problems: DPS (Chung et al., 2023), ΠGDM (Song et al., 2023a), TMPD (Boys et al., 2023), and two methods from Peng et al. (2024) - Peng (Convert) and Peng (Analytic). TMPD uses vector-Jacobian product $\mathbf{1}^\top \nabla_{\boldsymbol{x}_t} \boldsymbol{x}_0(\boldsymbol{s}_t)\sigma(t)^2$ for denoiser covariance. Convert employs neural network-output pixel-space diagonal covariance, while Analytic determines optimal constant pixel-diagonal covariances per timestep through moment matching. These

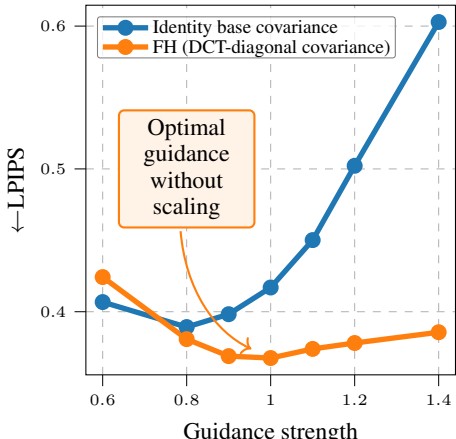

Figure 6: LPIPS w.r.t. guidance strength for the ImageNet validation set and the Gaussian blurring task. With a better covariance approximation, the usefulness of adjusting the approximated guidance $\nabla_{\boldsymbol{x}_t}$ with post-hoc tricks becomes smaller.

methods were selected as they represent reconstruction guidance with different covariances, enabling analysis of our covariance approximation approach. For DPS, we optimized guidance scale via ImageNet validation set sweeps. Non-identity covariance models used SciPy's conjugate gradient method for solving Eq. (23). For TMPD, we adjusted tolerance at higher noise levels to reduce generation time (see App. J).

**Scaling the guidance term** We investigated how covariance approximation affects the need for post-hoc changes to the estimated gradient $\nabla_{\boldsymbol{x}_t} \log p(\boldsymbol{y} \mid \boldsymbol{x}_t)$, as shown in Fig. 6. For deblurring, the cruder identity initialization required scaling slightly below 1, indicating an initial overestimation of the guidance scale. The more sophisticated DCT-diagonal covariance (FH) showed no systematic over- or underestimation, with optimal scaling at 1. We determined optimal guidance strength for identity covariance through a small sweep of 100 ImageNet validation samples at different solver step counts. No scaling was applied for DCT-diagonal covariance. Additional analysis with PSNR and SSIM is provided in App. K.

**Baseline comparisons** Our experiments focus on the low ODE sampling step regime to ensure practical applicability. Results in Table 1 show that adding online updates during sampling improves performance, with even greater gains when using DCT-diagonal covariance instead of the identity base covariance. On low step counts, our FH models consistently outperform baselines across all metrics, particularly on LPIPS scores. Visual comparisons in Fig. 7 and Fig. 9 confirm the effective fine detail preservation of FH at 15 and 30 steps. Extended results with 50 and 100 steps and the Euler solver in App. H, including comparisons to non-reconstruction guidance methods DDNM+ (Wang et al., 2023) and DiffPIR (Zhu et al., 2023), show FH and FH+Online almost always outperforming others at low step counts and typically achieving the best LPIPS scores even at higher step counts.

## 5 CONCLUSIONS

We introduced Free Hunch (FH), a framework for denoiser covariance estimation in diffusion models that leverages training data and trajectory curvature. FH provides accurate covariance estimates without additional training, architectural changes or ODE/SDE solver modifications. Our theoretical analysis showed that incorrect denoiser covariances significantly bias linear inverse problem solutions. Experiments on ImageNet demonstrated strong performance in linear inverse problems, especially at low step counts, with excellent LPIPS scores and fine detail preservation.

While FH introduces some additional complexity compared to simpler approaches, its efficiency and adaptability make it promising for various conditional generation tasks. Limitations of our work include the focus of the theoretical and experimental analysis on linear inverse problems, and future work could investigate nonlinear inverse problems and other types of conditional generation. Another open question is whether we can derive error bounds on the accuracy of the estimated covariance matrix in the entire process, or within individual 'time updates' or 'space updates'. While our DCT-diagonal base covariance works well for image data, the application to other data domains is another open question. A low-rank estimate of the covariance matrix with a PCA decomposition seems like a generally applicable approach, but this remains to be validated in practice.

Code for our approach is available at https://github.com/AaltoML/free-hunch.

Table 1: Comparison of image restoration methods for 15- and 30-step Heun iterations for deblurring (Gaussian), inpainting (random), deblurring (motion), and super-resolution ($4\times$) tasks. Our method (FH) excels overall, especially in the descriptive LPIPS metric. The top scores per category are **bolded**, with runners-up underlined. Close scores share rankings.

| | Method | Deblur (Gaussian) | | | Inpainting (Random) | | | Deblur (Motion) | | | Super res. ($4\times$) | | |
|---|---|---|---|---|---|---|---|---|---|---|---|---|---|
| | | PSNR↑ | SSIM↑ | LPIPS↓ | PSNR↑ | SSIM↑ | LPIPS↓ | PSNR↑ | SSIM↑ | LPIPS↓ | PSNR↑ | SSIM↑ | LPIPS↓ |
| 15 steps | DPS | 19.94 | 0.444 | 0.572 | 20.68 | 0.494 | 0.574 | 17.02 | 0.354 | 0.646 | 19.85 | 0.460 | 0.590 |
| | ΠGDM | 20.30 | 0.475 | 0.574 | 19.87 | 0.468 | 0.598 | 19.21 | 0.429 | 0.602 | 20.17 | 0.474 | 0.582 |
| | TMPD | 23.08 | 0.597 | 0.420 | 18.99 | 0.481 | 0.539 | 20.80 | 0.491 | 0.514 | 21.88 | 0.545 | 0.476 |
| | Peng (Convert) | 22.53 | 0.563 | 0.490 | 22.23 | 0.579 | 0.489 | 20.46 | 0.475 | 0.556 | 21.92 | 0.541 | 0.517 |
| | Peng (Analytic) | 22.53 | 0.563 | 0.490 | 22.14 | 0.574 | 0.494 | 20.46 | 0.475 | 0.556 | 21.92 | 0.541 | 0.517 |
| | Identity | 22.91 | 0.594 | 0.384 | 18.83 | 0.397 | 0.590 | 20.06 | 0.393 | 0.506 | 22.65 | 0.589 | 0.412 |
| | Identity+online | 23.08 | 0.606 | 0.385 | 18.86 | 0.397 | 0.590 | 20.31 | 0.418 | 0.492 | 22.76 | 0.597 | 0.414 |
| | FH | 23.39 | 0.624 | **0.372** | 24.73 | 0.701 | 0.327 | 21.69 | 0.534 | 0.446 | 23.30 | **0.624** | **0.390** |
| | FH+online | **23.54** | **0.634** | 0.378 | **25.25** | **0.728** | **0.317** | **21.84** | **0.549** | **0.441** | **23.39** | 0.632 | 0.394 |
| 30 steps | DPS | 21.76 | 0.527 | 0.463 | 24.84 | 0.678 | 0.387 | 18.22 | 0.389 | 0.582 | 23.00 | 0.593 | 0.440 |
| | ΠGDM | 22.27 | 0.559 | 0.468 | 21.24 | 0.518 | 0.517 | 21.16 | 0.508 | 0.503 | 22.11 | 0.553 | 0.478 |
| | TMPD | 23.16 | 0.602 | 0.415 | 18.85 | 0.481 | 0.537 | 20.91 | 0.500 | 0.507 | 21.94 | 0.549 | 0.472 |
| | Peng (Convert) | 23.61 | 0.627 | 0.405 | 23.74 | 0.648 | 0.403 | **21.99** | **0.553** | 0.463 | 23.22 | 0.608 | 0.430 |
| | Peng (Analytic) | 23.61 | 0.626 | 0.405 | 23.59 | 0.640 | 0.411 | 21.99 | **0.552** | 0.463 | 23.21 | 0.608 | 0.430 |
| | Identity | 23.15 | 0.602 | 0.374 | 18.75 | 0.402 | 0.578 | 20.14 | 0.406 | 0.494 | 22.82 | 0.588 | 0.405 |
| | Identity+online | 23.38 | 0.621 | 0.359 | 20.07 | 0.443 | 0.529 | 20.47 | 0.420 | 0.467 | 23.38 | 0.622 | 0.383 |
| | FH | 23.55 | 0.630 | **0.353** | 26.00 | 0.757 | **0.256** | 21.80 | 0.538 | **0.411** | 23.38 | 0.623 | **0.372** |
| | FH+online | **23.62** | **0.635** | 0.358 | **26.18** | **0.767** | 0.268 | 21.88 | 0.547 | 0.410 | **23.44** | **0.628** | 0.375 |

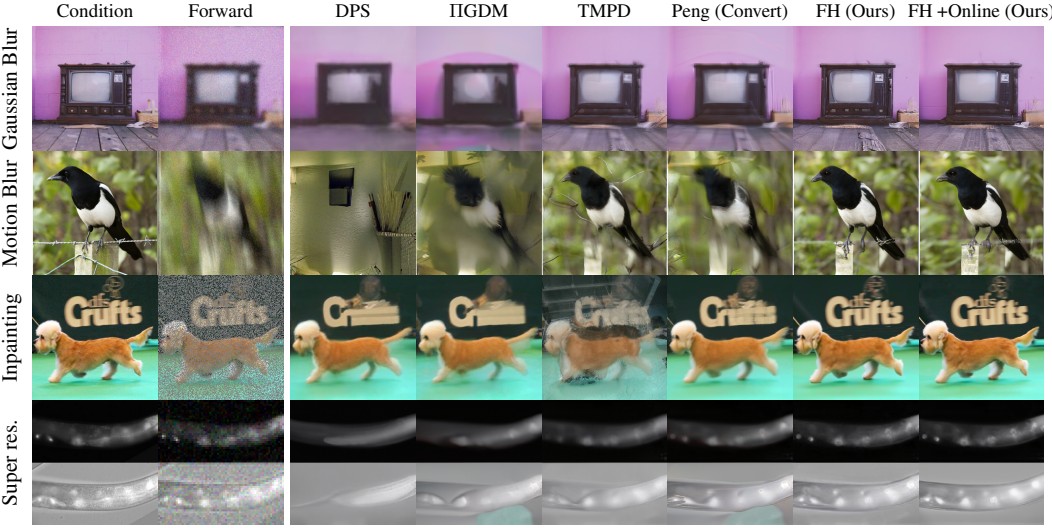

Figure 7: Qualitative examples using the 15-step Heun sampler for image restoration methods for deblurring (Gaussian), inpainting (random), deblurring (motion), and super-resolution ($4\times$) tasks. Quantitative metrics in Table 1. Our method manages to restore the corrupted ('Forward') to match well with the original ('Condition').

ACKNOWLEDGMENTS

We acknowledge funding from the Research Council of Finland (grants 339730, 362408, 334600). We acknowledge CSC – IT Center for Science, Finland, for awarding this project access to the LUMI supercomputer, owned by the EuroHPC Joint Undertaking, hosted by CSC (Finland) and the LUMI consortium through CSC. We acknowledge the computational resources provided by the Aalto Science-IT project.

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

APPENDICES

## A  FULL DERIVATIONS FOR THE TIME UPDATE

For this section, we denote $p(\boldsymbol{x}_t) = \log p(\boldsymbol{x}, t)$ to explicitly separate the time variable from the spatial variable. This notation is useful for the derivations below, but in other parts of the paper we use the $\boldsymbol{x}_t$ to separate it from $\boldsymbol{x}_0$.

Recall the mean and covariance of the Gaussian approximation at location $\boldsymbol{x}$ and time $t$:

$$\boldsymbol{m}(\boldsymbol{x}, t) = \boldsymbol{x} - \nabla_{\boldsymbol{x}}^2 \log p(\boldsymbol{x}, t)^{-1} \nabla_{\boldsymbol{x}} \log p(\boldsymbol{x}, t), \tag{26}$$

$$\boldsymbol{C}(\boldsymbol{x}, t) = -\nabla_{\boldsymbol{x}}^2 \log p(\boldsymbol{x}, t)^{-1}. \tag{27}$$

The evolution of the Gaussian has a closed form (Särkkä & Solin, 2019). In the variance-exploding case this results in

$$\boldsymbol{m}(\boldsymbol{x}, t + \Delta t) = \boldsymbol{m}(\boldsymbol{x}, t), \tag{28}$$

$$\boldsymbol{C}(\boldsymbol{x}, t + \Delta t) = \boldsymbol{C}(\boldsymbol{x}, t) + \Delta\sigma^2 \boldsymbol{I}, \tag{29}$$

where $\Delta\sigma^2 = \sigma^2(t + \Delta t) - \sigma^2(t)$. That is, the forward keeps the mean intact while increasing the covariance. Using the above equations we can derive:

$$\nabla_{\boldsymbol{x}}^2 \log p(\boldsymbol{x}, t + \Delta t) = \left(\nabla_{\boldsymbol{x}}^2 \log p(\boldsymbol{x}, t)^{-1} - \Delta\sigma^2 \boldsymbol{I}\right)^{-1} \tag{30}$$

$$\nabla_{\boldsymbol{x}} \log p(\boldsymbol{x}, t + \Delta t) = \nabla_{\boldsymbol{x}}^2 \log p(\boldsymbol{x}, t + \Delta t) \nabla_{\boldsymbol{x}}^2 \log p(\boldsymbol{x}, t)^{-1} \nabla_{\boldsymbol{x}} \log p(\boldsymbol{x}, t) \tag{31}$$

When connected with Equation (7) and Equation (8), we can now derive the denoiser mean and covariance updates.

$$\boldsymbol{\mu}_{0\,|\,t+\Delta t}(\boldsymbol{x}) = \boldsymbol{x} + \sigma^2(t + \Delta t) \nabla_{\boldsymbol{x}} \log p(\boldsymbol{x}, t + \Delta t)$$

$$= \boldsymbol{x} + \sigma^2(t + \Delta t) \underbrace{\left(\nabla^2 \log p(\boldsymbol{x}, t)^{-1} - \Delta\sigma^2 I\right)^{-1}}_{\text{Hessian projection}} \underbrace{\nabla^2 \log p(\boldsymbol{x}, t)^{-1} \nabla_{\boldsymbol{x}} \log p(\boldsymbol{x}, t)}_{\text{Hessian-score product}} \tag{32}$$

$$\boldsymbol{\mu}_{0\,|\,t+\Delta t}(\boldsymbol{x}) = \boldsymbol{x} + \sigma(t + \Delta t)^2 \left(\left(\sigma(t)^2 + \Delta\sigma^2\right)\boldsymbol{I} - \frac{\Delta\sigma^2}{\sigma(t)^2}\boldsymbol{\Sigma}_{0\,|\,t}(\boldsymbol{x})\right)^{-1} \left(\boldsymbol{\mu}_{0\,|\,t}(\boldsymbol{x}) - \boldsymbol{x}\right),$$

$$= \boldsymbol{x} + \sigma(t + \Delta t)^2 \left(\sigma(t + \Delta t)^2 \boldsymbol{I} - \frac{\Delta\sigma^2}{\sigma(t)^2}\boldsymbol{\Sigma}_{0\,|\,t}(\boldsymbol{x})\right)^{-1} \left(\boldsymbol{\mu}_{0\,|\,t}(\boldsymbol{x}) - \boldsymbol{x}\right), \tag{33}$$

$$\boldsymbol{\Sigma}_{0\,|\,t+\Delta t}(\boldsymbol{x}) = \left(\boldsymbol{\Sigma}_{0\,|\,t}(\boldsymbol{x})^{-1} + \Delta\sigma^{-2}\boldsymbol{I}\right)^{-1}. \tag{34}$$

This result is not entirely obvious, and next we will provide a detailed derivation.

**Deriving the denoiser covariance update**  With the locally Gaussian approximation on $p(\boldsymbol{x}, t)$, we can represent the time evolution of the $p(\boldsymbol{x}, t)$ covariance as the following

$$\boldsymbol{C}(\boldsymbol{x}, t) = \boldsymbol{C}_0 + \sigma(t)^2 \boldsymbol{I}, \tag{35}$$

where $\boldsymbol{C}_0$ is the hypothetical covariance when extrapolating the Gaussian time evolution to $t = 0$. Then, moving back to the $\boldsymbol{x}_t$ notation, we have the following connections (Eq. (11))

$$\nabla_{\boldsymbol{x}_t}^2 \log p(\boldsymbol{x}_t) = -(\boldsymbol{C}_0 + \sigma(t)^2 \boldsymbol{I})^{-1}, \tag{36}$$

$$\nabla_{\boldsymbol{x}_t}^2 \log p(\boldsymbol{x}_t)^{-1} = -(\boldsymbol{C}_0 + \sigma(t)^2 \boldsymbol{I}). \tag{37}$$

On the other hand, the denoiser covariance is

$$\mathbb{C}\text{ov}[\boldsymbol{x}_0 \,|\, \boldsymbol{x}_t] = (\nabla_{\boldsymbol{x}_t}^2 \log p(\boldsymbol{x}_t)\sigma(t)^2 + \boldsymbol{I})\sigma(t)^2. \tag{38}$$

The inverse of the denoiser covariance is then (Sherman–Morrison–Woodbury formula):

$$\mathbb{C}\text{ov}[\boldsymbol{x}_0 \,|\, \boldsymbol{x}_t]^{-1} = (\nabla_{\boldsymbol{x}_t}^2 \log p(\boldsymbol{x}_t)\sigma(t)^2 + \boldsymbol{I})^{-1}\sigma(t)^{-2} \tag{39}$$

$$= (\boldsymbol{I} - (\boldsymbol{I} + \nabla_{\boldsymbol{x}_t}^2 \log p(\boldsymbol{x}_t)^{-1}\sigma(t)^{-2})^{-1})\sigma(t)^{-2} \quad \text{(Woodbury)} \tag{40}$$

$$= \left(\boldsymbol{I} - \left(\boldsymbol{I} - (\boldsymbol{C}_0 + \sigma(t)^2\boldsymbol{I})\sigma(t)^{-2}\right)^{-1}\right)\sigma(t)^{-2} \quad \text{(37)} \tag{41}$$

$$= (\boldsymbol{I} + \boldsymbol{C}_0^{-1}\sigma(t)^2)\sigma(t)^{-2} \tag{42}$$

$$= \boldsymbol{C}_0^{-1} + \sigma(t)^{-2}\boldsymbol{I}. \tag{43}$$

So the inverse of the denoiser covariance is simply a constant term plus an identity scaled with $\sigma(t)^{-2}$. This means that

$$\mathbb{C}\mathrm{ov}[\boldsymbol{x}_0 \,|\, \boldsymbol{x}_{t+\Delta t}]^{-1} - \mathbb{C}\mathrm{ov}[\boldsymbol{x}_0 \,|\, \boldsymbol{x}_t]^{-1} = (\boldsymbol{C}_0^{-1} + \boldsymbol{I}\sigma(t+\Delta t)^{-2}) - (\boldsymbol{C}_0^{-1} + \boldsymbol{I}\sigma(t)^{-2}) \tag{44}$$

$$\mathbb{C}\mathrm{ov}[\boldsymbol{x}_0 \,|\, \boldsymbol{x}_{t+\Delta t}]^{-1} = \mathbb{C}\mathrm{ov}[\boldsymbol{x}_0 \,|\, \boldsymbol{x}_t]^{-1} + \boldsymbol{I}\sigma(t+\Delta t)^{-2} - \boldsymbol{I}\sigma(t)^{-2} \tag{45}$$

$$= \mathbb{C}\mathrm{ov}[\boldsymbol{x}_0 \,|\, \boldsymbol{x}_t]^{-1} + \Delta\sigma(t)^{-2}\boldsymbol{I}. \tag{46}$$

And thus

$$\mathbb{C}\mathrm{ov}[\boldsymbol{x}_0 \,|\, \boldsymbol{x}_{t+\Delta t}] = (\mathbb{C}\mathrm{ov}[\boldsymbol{x}_0 \,|\, \boldsymbol{x}_t]^{-1} + \Delta\sigma(t)^{-2}\boldsymbol{I})^{-1}. \tag{47}$$

**Deriving the denoiser mean update** We want to calculate the expression for the updated mean $\boldsymbol{\mu}_{0\,|\,t+\Delta t}(\boldsymbol{x}_t)$. This mean can be written as:

$$\boldsymbol{\mu}_{0\,|\,t+\Delta t}(\boldsymbol{x}_t) = \boldsymbol{x}_t + \sigma(t+\Delta t)^2 \cdot \left(\nabla^2_{\boldsymbol{x}_t} \log p(\boldsymbol{x}_t)^{-1} - \Delta\sigma^2\boldsymbol{I}\right)^{-1} \nabla^2_{\boldsymbol{x}_t} \log p(\boldsymbol{x}_t)^{-1} \nabla_{\boldsymbol{x}_t} \log p(\boldsymbol{x}_t). \tag{48}$$

We aim to simplify the term inside the parentheses. Starting from the observation:

$$\left(\nabla^2_{\boldsymbol{x}_t} \log p(\boldsymbol{x}_t)^{-1} - \Delta\sigma^2\boldsymbol{I}\right)^{-1} \nabla^2_{\boldsymbol{x}_t} \log p(\boldsymbol{x}_t)^{-1} = \left(\boldsymbol{I} - \Delta\sigma^2\nabla^2_{\boldsymbol{x}_t} \log p(\boldsymbol{x}_t)\right)^{-1} \tag{49}$$

and using Eq. (7) and Eq. (8), we can express $\nabla^2_{\boldsymbol{x}_t} \log p(\boldsymbol{x}_t)$ and $\nabla_{\boldsymbol{x}_t} \log p(\boldsymbol{x}_t)$ as functions of $\boldsymbol{\Sigma}_{0\,|\,t}(\boldsymbol{x}_t)$ and $\sigma(t)$, yielding:

$$\boldsymbol{\mu}_{0\,|\,t+\Delta t}(\boldsymbol{x}_t) = \boldsymbol{x}_t + \sigma(t+\Delta t)^2 \cdot \left(\left(\sigma(t)^2 + \Delta\sigma^2\right)\boldsymbol{I} - \frac{\Delta\sigma^2}{\sigma(t)^2}\boldsymbol{\Sigma}_{0\,|\,t}(\boldsymbol{x}_t)\right)^{-1} \left(\boldsymbol{\mu}_{0\,|\,t}(\boldsymbol{x}_t) - \boldsymbol{x}_t\right). \tag{50}$$

This is the final simplified expression for $\boldsymbol{\mu}_{0\,|\,t+\Delta t}(\boldsymbol{x}_t)$.

## B INVERTING THE EFFICIENT MATRIX REPRESENTATION

Let's say we have a positive-definite matrix in represented in the format $\boldsymbol{C} = \boldsymbol{D} + \boldsymbol{U}\boldsymbol{U}^\top - \boldsymbol{V}\boldsymbol{V}^\top$, where $\boldsymbol{D}$ is a diagonal matrix and $U$ and $V$ are $N \times k_1$ and $N \times k_2$ matrices, respectively. $N$ is the data dimensionality and $k \ll N$. To invert it, we use the Woodbury identity twice, first for $\boldsymbol{A} = \boldsymbol{D} + \boldsymbol{U}\boldsymbol{U}^\top$, and second for $\boldsymbol{A} - \boldsymbol{V}\boldsymbol{V}^\top$. The first application of the identity is:

$$\boldsymbol{A}^{-1} = (\boldsymbol{D} + \boldsymbol{U}\boldsymbol{U}^\top)^{-1} = \boldsymbol{D}^{-1} - \boldsymbol{D}^{-1}\boldsymbol{U} \underbrace{(\boldsymbol{I} + \boldsymbol{U}^\top\boldsymbol{D}^{-1}\boldsymbol{U})^{-1}}_{=\boldsymbol{K}} \boldsymbol{U}^\top\boldsymbol{D}^{-1} \tag{51}$$

$$= \boldsymbol{D}^{-1} - \underbrace{\boldsymbol{D}^{-1}\boldsymbol{U}\,\mathrm{sqrt}(\boldsymbol{K})}_{=\boldsymbol{V}'} \mathrm{sqrt}(\boldsymbol{K})^\top\boldsymbol{U}^\top\boldsymbol{D}^{-1} \tag{52}$$

$$= \boldsymbol{D}^{-1} - \boldsymbol{V}'\boldsymbol{V}'^\top \tag{53}$$

that is, when we invert $\boldsymbol{A} = \boldsymbol{D} + \boldsymbol{U}\boldsymbol{U}^\top$, we get something in the form $\boldsymbol{D}^{-1} - \boldsymbol{V}'\boldsymbol{V}'^\top$. Note that $\boldsymbol{I} - \boldsymbol{U}^\top\boldsymbol{D}^{-1}\boldsymbol{U}$ is a $k_1 \times k_1$ matrix, instea of an $N \times N$ matrix and as such is much more efficient to invert than the full $N \times N$ matrix when $k_1 \ll N$. Now, invert $\boldsymbol{C} = \boldsymbol{A} - \boldsymbol{V}\boldsymbol{V}^\top$:

$$\boldsymbol{C}^{-1} = (\boldsymbol{A} - \boldsymbol{V}\boldsymbol{V}^\top)^{-1} = \boldsymbol{A}^{-1} + \boldsymbol{A}^{-1}\boldsymbol{V} \underbrace{(\boldsymbol{I} - \boldsymbol{V}^\top\boldsymbol{A}^{-1}\boldsymbol{V})^{-1}}_{=\boldsymbol{L}} \boldsymbol{V}^\top\boldsymbol{A}^{-1} \tag{54}$$

$$= \boldsymbol{A}^{-1} + \underbrace{\boldsymbol{A}^{-1}\boldsymbol{V}\,\mathrm{sqrt}(\boldsymbol{L})}_{=\boldsymbol{U}'} \mathrm{sqrt}(\boldsymbol{K})^\top\boldsymbol{V}^\top\boldsymbol{A}^{-1} \tag{55}$$

$$= \boldsymbol{A}^{-1} + \boldsymbol{U}'\boldsymbol{U}' = \boldsymbol{D}^{-1} + \boldsymbol{U}'\boldsymbol{U}'^\top - \boldsymbol{V}'\boldsymbol{V}'^\top. \tag{56}$$

Note that $\boldsymbol{V}^\top\boldsymbol{A}^{-1}\boldsymbol{V}$ is again efficient to compute due to the low-rank structure of $\boldsymbol{A}^{-1}$:

$$\boldsymbol{V}^\top\boldsymbol{A}^{-1}\boldsymbol{V} = \boldsymbol{V}^\top \left(\boldsymbol{D}^{-1} - \boldsymbol{V}'\boldsymbol{V}'^\top\right) \boldsymbol{V} \tag{57}$$

$$= \boldsymbol{V}^\top\boldsymbol{D}^{-1}\boldsymbol{V} - \boldsymbol{V}^\top\boldsymbol{V}'\boldsymbol{V}'^\top\boldsymbol{V} \tag{58}$$

$$= \underbrace{\boldsymbol{V}^\top\boldsymbol{D}^{-1}\boldsymbol{V}}_{k_2 \times k_2} - \underbrace{(\boldsymbol{V}^\top\boldsymbol{V}')}_{k_2 \times k_2}\underbrace{(\boldsymbol{V}'^\top\boldsymbol{V})}_{k_2 \times k_2}. \tag{59}$$

This leads to a well-behaved $k_2 \times k_2$ matrix, and the inverse $(\boldsymbol{I} - \boldsymbol{V}^\top \boldsymbol{A}^{-1} \boldsymbol{V})^{-1}$ is also efficient to compute.

**The matrix square root and complex numbers** Note that in addition to the $k_1 \times k_1$ and $k_2 \times k_2$ inverses, the method requires matrix square root. One might imagine that $(\boldsymbol{I} + \boldsymbol{U}^\top \boldsymbol{D}^{-1} \boldsymbol{U})^{-1}$ is guaranteed to be positive-definite due to the original matrix being such, but this is not necessarily the case. $(\boldsymbol{I} + \boldsymbol{U}^\top \boldsymbol{U})^{-1}$ would be, but the diagonal term $\boldsymbol{D}^{-1}$ can push the eigenvalues of the matrix to negative. This means that we can not use the Cholesky decomposition for the matrix square root operations, but instead we use the Schur decomposition as implemented in the scipy library. A side-effect is also that we have to use complex numbers to represent $\boldsymbol{D}$, $\boldsymbol{U}$, and $\boldsymbol{V}$ in our implementation. This is not an issue, since in the calculation of the covariance $\boldsymbol{D} + \boldsymbol{U}\boldsymbol{U}^\top - \boldsymbol{V}\boldsymbol{V}^\top$, the imaginary components cancel out and we get a real matrix.

## C  EXTENDED ANALYSIS OF THE TOY EXAMPLE

As a recap, in the toy model, $\boldsymbol{A} = \boldsymbol{I}$ and all the pixels are perfectly correlated with $\mathbb{Cov}[\boldsymbol{x}_0] = \boldsymbol{J}$, where $\boldsymbol{J}$ is a matrix full of ones. The observation $\boldsymbol{y}$ and the denoiser mean $\boldsymbol{\mu}_{0\,|\,t}(\boldsymbol{x}_t)$ are also vectors of ones $\vec{\boldsymbol{1}}$ scaled by a constant, so that $\boldsymbol{y} - \boldsymbol{\mu}_{0\,|\,t}(\boldsymbol{x}_t) = a\vec{\boldsymbol{1}}$.

**ΠGDM guidance without postprocessing**

$$(\boldsymbol{y} - \boldsymbol{\mu}_{0\,|\,t}(\boldsymbol{x}_t))^\top (\boldsymbol{I}\frac{\sigma(t)^2}{1+\sigma(t)^2} + \sigma_y^2\boldsymbol{I})^{-1}\nabla_{\boldsymbol{x}_t}\boldsymbol{\mu}_{0\,|\,t}(\boldsymbol{x}_t)$$

$$\approx (\boldsymbol{y} - \boldsymbol{\mu}_{0\,|\,t}(\boldsymbol{x}_t))^\top (\boldsymbol{I} + \sigma_y^2\boldsymbol{I})^{-1}\frac{\mathbb{Cov}[\boldsymbol{x}_0\,|\,\boldsymbol{x}_t]}{\sigma(t)^2} \tag{60}$$

$$= \frac{a}{1+\sigma_y^2}\vec{\boldsymbol{1}}^\top\frac{\boldsymbol{J}}{\sigma(t)^2} = \frac{aN}{(1+\sigma_y^2)\sigma(t)^2}\vec{\boldsymbol{1}}^\top. \tag{61}$$

**DPS guidance without postprocessing**

$$(\boldsymbol{y} - \boldsymbol{\mu}_{0\,|\,t}(\boldsymbol{x}_t))^\top (\boldsymbol{I}0 + \sigma_y^2\boldsymbol{I})^{-1}\nabla_{\boldsymbol{x}_t}\boldsymbol{\mu}_{0\,|\,t}(\boldsymbol{x}_t)$$

$$\approx (\boldsymbol{y} - \boldsymbol{\mu}_{0\,|\,t}(\boldsymbol{x}_t))^\top\sigma_y^{-2}\frac{\mathbb{Cov}[\boldsymbol{x}_0\,|\,\boldsymbol{x}_t]}{\sigma(t)^2} \tag{62}$$

$$= \frac{a}{\sigma_y^2}\vec{\boldsymbol{1}}^\top\frac{\boldsymbol{J}}{\sigma(t)^2} = \frac{aN}{\sigma_y^2\sigma(t)^2}\vec{\boldsymbol{1}}^\top. \tag{63}$$

Here $N$ is the data dimensionality.

For ΠGDM, the gradient is scaled by $\frac{\sigma(t)^2}{1+\sigma(t)^2}$, but this does not change the result in high noise levels. Instead, the clipping of the denoiser mean to $[-1, 1]$ regularises the guidance such that the generation trajectory does not blow up. For DPS, the additional scaling results in

$$\sigma(t)^2\nabla_{\boldsymbol{x}_t}p(\boldsymbol{y}\,|\,\boldsymbol{x}_t) \approx \frac{\xi\sigma_y^2}{\|\boldsymbol{y} - \boldsymbol{A}\boldsymbol{x}_0\|}\frac{N}{\sigma_y^2}\vec{\boldsymbol{1}} = \frac{\xi N}{\|\vec{\boldsymbol{1}}\|}\vec{\boldsymbol{1}} = \frac{\xi N}{\sqrt{N}}\vec{\boldsymbol{1}} = \xi\sqrt{N}\vec{\boldsymbol{1}} \tag{64}$$

which is less severe than ΠGDM, but still requires additional clipping unless the scale $\xi$ is set to very low values.

**Solution with the correct covariance** In the main text, we showed that the issue does not show up in the case $\sigma_y = 0$. This resulted in:

$$\nabla_{\boldsymbol{x}_t}\log p(\boldsymbol{y}\,|\,\boldsymbol{x}_t) = (\boldsymbol{y} - \boldsymbol{x}_0(\boldsymbol{x}_t))^\top\mathbb{Cov}[\boldsymbol{x}_0\,|\,\boldsymbol{x}_t]^{-1}\frac{\mathbb{Cov}[\boldsymbol{x}_0\,|\,\boldsymbol{x}_t]}{\sigma_t^2} = \frac{(\boldsymbol{y} - \boldsymbol{x}_0(\boldsymbol{x}_t))^\top}{\sigma_t^2} \tag{65}$$

The corresponding $\boldsymbol{x}_0$ estimate is:

$$\boldsymbol{x}_t + \sigma_t^2\left(\nabla_{\boldsymbol{x}_t}\log p(\boldsymbol{x}_t) + \frac{(\boldsymbol{y} - \boldsymbol{x}_0(\boldsymbol{x}_t))}{\sigma_t^2}\right)$$

$$= \mathbb{E}[\boldsymbol{x}_0\,|\,\boldsymbol{x}_t] + \boldsymbol{y} - \mathbb{E}[\boldsymbol{x}_0\,|\,\boldsymbol{x}_t] = \boldsymbol{y} \tag{66}$$

that is, the updated score points to the direction of the observation for all time steps $t$. For the case $t \to \infty$ and $\mathbb{C}\text{ov}[\boldsymbol{x}_0 \,|\, \boldsymbol{x}_t] \approx \boldsymbol{J}$ and **not** assuming $\sigma_y = 0$, we can derive with the Sherman–Morrison formula that

$$(\mathbb{C}\text{ov}[\boldsymbol{x}_0 \,|\, \boldsymbol{x}_t] + \sigma_y^2 \boldsymbol{I})^{-1} = (J + \sigma_y^2 I)^{-1} = \frac{1}{\sigma_y^2}\boldsymbol{I} - \frac{1}{\sigma_y^4 + N\sigma_y^2}\boldsymbol{J}. \tag{67}$$

To simplify formulas, let us again assume that $\boldsymbol{y} - \boldsymbol{x}_0(\boldsymbol{x}_t) = a\vec{\boldsymbol{1}}$

$$\nabla_{\boldsymbol{x}_t} \log p(\boldsymbol{y} \,|\, \boldsymbol{x}_t) \approx a\vec{\boldsymbol{1}}^\top \left( \frac{1}{\sigma_y^2}\boldsymbol{I} - \frac{1}{\sigma_y^4 + N\sigma_y^2}\boldsymbol{J} \right) \frac{\boldsymbol{J}}{\sigma_t^2} \tag{68}$$

$$= a\left( \frac{1}{\sigma_y^2} - \frac{N}{\sigma_y^4 + N\sigma_y^2} \right) \vec{\boldsymbol{1}}^\top \frac{\boldsymbol{J}}{\sigma_t^2} \tag{69}$$

$$= a\frac{1}{\sigma_y^2 + N} \vec{\boldsymbol{1}}^\top \frac{\boldsymbol{J}}{\sigma_t^2} \tag{70}$$

$$= a\frac{1}{\sigma_y^2 + N} \frac{N}{\sigma_t^2} \vec{\boldsymbol{1}}^\top. \tag{71}$$

Here, again, since the inverse term is inversely dependent on $N$, the dependence of the last term on $N$ is cancelled. In the case $\sigma_y^2 = 0$, we recover the exact same result as previously. With non-zero observation noise, the strength of the guidance becomes slightly smaller, reflecting the uncertainty about the underlying pure $\boldsymbol{x}_0$ value we have measured.

## D  FULL GUIDANCE ALGORITHMS FOR THE LINEAR-GAUSSIAN OBSERVATION MODEL

The algorithm Alg. 3 details an implementation of the method with the Euler ODE solver. The algorithm Alg. 4 is a more easily applicable implementation with any type of solver, including higher-order methods like the Heun method. In it, we instantiate a class at the beginning of sampling, and whenever a call to the denoiser / score model is made, it is passed to the class to calculate $\nabla_{\boldsymbol{x}_t} \log p(\boldsymbol{x}_t)$ and update the covariance information. For image data, we only perform the space updates in $1 < \sigma(t) < 5$, as detailed in App. J.

## E  ADDITIONAL TOY EXPERIMENT WITH CORRELATED DATA

To examine the effect mentioned in Sec. 3.7, we constructed data $p(\boldsymbol{x}_0) = \mathcal{N}(\boldsymbol{x}_0 \,|\, \boldsymbol{0}, (1-\rho)\boldsymbol{I} + \rho\boldsymbol{J})$, where $\boldsymbol{J}$ is a matrix of ones and $\rho = 0.999$. We used an observation with noise $\sigma_y = 0.2$ and varied the dimension. We plot the variance of the generated samples in Fig. 8. As expected, both $\Pi$GDM and DPS become overly confident as dimensionality increases. In contrast, our method, which explicitly accounts for data covariance, maintains correct uncertainty calibration across dimension counts. Note that the DPS results are obtained after tuning the guidance scale for this particular problem, making the comparisons somewhat favourable towards DPS.

## F  ADDITIONAL QUALITATIVE RESULTS

Figure 9 shows the qualitative comparison with 30 Heun solver steps.

## G  QUANTIFYING THE ERROR IN THE COVARIANCE ESTIMATION

We study the case where we directly estimate the error in the denoiser covariance estimates for low-dimensional toy data, where this is directly feasible. We consider a 2D Gaussian mixture model with an likelihood function and posterior as shown in Fig. 10.

We generate samples with four different methods, and compare the covariances true to the true covariance with the Frobenius norm. The methods are:

---

**Algorithm 3:** Free Hunch for Linear Inverse Problems (Euler solver, with diffusion parameters from (Karras et al., 2022))

---

**Input:** Linear operator $\boldsymbol{A}$, observation $\boldsymbol{y}$, noise $\sigma_y$
**Input:** Initial covariance $\boldsymbol{\Sigma}_{\text{data}}$, score model $s_\theta$
**Input:** Schedule params: $\sigma_{\min} = t_{\min} = 0.002$, $\sigma_{\max} = t_{\max} = 80$, $\rho = 7$, steps $N$

   /* Define time discretization                                      */

1   $t_i = (t_{\max}^{\frac{1}{\rho}} + \frac{i}{N-1} t_{\min}^{\frac{1}{\rho}} - t_{\max}^{\frac{1}{\rho}})^\rho$ for $i < N$, $t_N = 0$ (Karras et al., 2022)

2   Initialize $\boldsymbol{x}_t \sim \mathcal{N}(\boldsymbol{0}, \sigma_{\max}^2 \boldsymbol{I})$

3   Initialize $\boldsymbol{\Sigma}_{0\,|\,t}(\boldsymbol{x}_t) = \boldsymbol{\Sigma}_{\text{data}}$

4   Initialize $\boldsymbol{\mu}_{0\,|\,t_i}^{\text{transferred}} = null$

5   Initialize $\Delta \boldsymbol{x} = null$

6   **for** $i = 1, \ldots, N-1$ **do**

7      $\sigma_{\text{curr}} = t_i$, $\sigma_{\text{next}} = t_{i+1}$

8      $\Delta t = t_{i+1} - t_i$

      /* New score and denoising mean evaluation                    */

9      $\nabla_{\boldsymbol{x}_t} \log p(\boldsymbol{x}_t) = s_\theta(\boldsymbol{x}_t, t_i)$

10     $\boldsymbol{\mu}_{0\,|\,t}(\boldsymbol{x}_t) = \boldsymbol{x}_t + \sigma_{\text{curr}}^2 \nabla_{\boldsymbol{x}_t} \log p(\boldsymbol{x}_t)$

      /* Space update for covariance                                */

11     **if** $\boldsymbol{\mu}_{0\,|\,t_i}^{\text{transferred}} \neq null$ *And* $\Delta \boldsymbol{x} \neq null$ **then**

12        $\Delta \boldsymbol{e} = \sigma_{\text{curr}}^2 (\boldsymbol{\mu}_{0\,|\,t}(\boldsymbol{x}_t) - \boldsymbol{\mu}_{0\,|\,t_i}^{\text{transferred}})$

13        $\gamma = \frac{1}{\Delta \boldsymbol{e}^\top \Delta \boldsymbol{x}}$

14        $\boldsymbol{\Sigma}_{0\,|\,t}(\boldsymbol{x}_{t_{\text{next}}}) = \boldsymbol{\Sigma}_{0\,|\,t}(\boldsymbol{x}_t) - \frac{\boldsymbol{\Sigma}_{0\,|\,t}(\boldsymbol{x}_t) \Delta \boldsymbol{x} \Delta \boldsymbol{x}^\top \boldsymbol{\Sigma}_{0\,|\,t}(\boldsymbol{x}_t)}{\Delta \boldsymbol{x}^\top \boldsymbol{\Sigma}_{0\,|\,t}(\boldsymbol{x}_t) \Delta \boldsymbol{x}} + \frac{\Delta \boldsymbol{e} \Delta \boldsymbol{e}^\top}{\Delta \boldsymbol{e}^\top \Delta \boldsymbol{x}}$

15     **end**

      /* Reconstruction guidance                                      */

16     $\nabla_{\boldsymbol{x}_t} \log p(\boldsymbol{y}|\boldsymbol{x}_t) = (\boldsymbol{y} - \boldsymbol{A}\boldsymbol{\mu}_{0\,|\,t}(\boldsymbol{x}_t))^\top (\boldsymbol{A}\boldsymbol{\Sigma}_{0\,|\,t}(\boldsymbol{x}_t)\boldsymbol{A}^\top + \sigma_y^2 \boldsymbol{I})^{-1} \boldsymbol{A} \nabla_{\boldsymbol{x}_t} \boldsymbol{\mu}_{0\,|\,t}(\boldsymbol{x}_t)$

      /* Fall back to approximation if guidance too large       */

17     **if** $\|\sigma_{curr}^2 \nabla_{\boldsymbol{x}_t} \log p(\boldsymbol{y}\,|\,\boldsymbol{x}_t)\| > 1$ **then**

18        $\nabla_{\boldsymbol{x}_t} \log p(\boldsymbol{y}|\boldsymbol{x}_t) = (\boldsymbol{y} - \boldsymbol{A}\boldsymbol{\mu}_{0\,|\,t}(\boldsymbol{x}_t))^\top (\boldsymbol{A}\boldsymbol{\Sigma}_{0\,|\,t}(\boldsymbol{x}_t)\boldsymbol{A}^\top + \sigma_y^2 \boldsymbol{I})^{-1} \boldsymbol{A} \frac{\boldsymbol{\Sigma}_{0\,|\,t}(\boldsymbol{x}_t)}{\sigma_{\text{curr}}^2}$

19     **end**

      /* Update sample with Euler step                              */

20     $\Delta \boldsymbol{x} = -\sigma_{\text{curr}}(\nabla_{\boldsymbol{x}_t} \log p(\boldsymbol{x}_t) + \nabla_{\boldsymbol{x}_t} \log p(\boldsymbol{y}|\boldsymbol{x}_t))\Delta t$

21     $\boldsymbol{x}_t = \boldsymbol{x}_t + \Delta x$

      /* Time update for mean                                      */

22     $\Delta \sigma^2 = \sigma_{\text{next}}^2 - \sigma_{\text{curr}}^2$

23     $\boldsymbol{\mu}_{0\,|\,t_{i+1}}^{\text{transferred}} = \boldsymbol{x}_t + \sigma_{\text{next}}^2 (\sigma_{\text{next}}^2 \boldsymbol{I} - \frac{\Delta \sigma^2}{\sigma_{\text{curr}}^2} \boldsymbol{\Sigma}_{0\,|\,t}(\boldsymbol{x}_t))^{-1}(\boldsymbol{\mu}_{0\,|\,t}(\boldsymbol{x}_t) - \boldsymbol{x}_t)$

      /* Time update for covariance (moving to noise level $\sigma_{\text{next}}$) */

24     $\Delta(\sigma^{-2}) = \sigma_{\text{next}}^{-2} - \sigma_{\text{curr}}^{-2}$

25     $\boldsymbol{\Sigma}_{0\,|\,t}(\boldsymbol{x}_t)^{-1} = \boldsymbol{\Sigma}_{0\,|\,t}(\boldsymbol{x}_t)^{-1} + \Delta(\sigma^{-2})\boldsymbol{I}$

26     $\boldsymbol{\Sigma}_{0\,|\,t}(\boldsymbol{x}_t) = (\boldsymbol{\Sigma}_{0\,|\,t}(\boldsymbol{x}_t)^{-1})^{-1}$

27 **end**

28 **return** $\boldsymbol{x}_t$

---

---

**Algorithm 4:** Free Hunch Guidance Class, applicable with any solver

---

1 **class** FreeHunchGuidance:

/* Initialize with measurement model and data covariance */

2     **constructor**($\boldsymbol{A}$, $\boldsymbol{y}$, $\sigma_y$, $\boldsymbol{\Sigma}_{\text{data}}$):

3         Store $\boldsymbol{A}$, $\boldsymbol{y}$, $\sigma_y$, $\boldsymbol{\Sigma}_{0\,|\,t} = \boldsymbol{\Sigma}_{\text{data}}$

4         Initialize $\boldsymbol{\mu}_{\text{prev}} = null$, $\boldsymbol{x}_{\text{prev}} = null$, $\sigma_{\text{prev}} = null$

/* Process new denoiser evaluation and return guidance, to be used for updating $\nabla_{\boldsymbol{x}_t} \log p(\boldsymbol{x}_t)$ to $\nabla_{\boldsymbol{x}_t} \log p(\boldsymbol{x}_t) + \nabla_{x_t} \log p(\boldsymbol{y}\,|\,\boldsymbol{x}_t)$ before using it in the solver. */

5     **function** process_denoiser($\boldsymbol{\mu}_{\text{new}}$, $\boldsymbol{x}_{\text{new}}$, $\sigma_{\text{new}}$):

6         **if** $\boldsymbol{\mu}_{prev} \neq null$ **then**

/* Time update from previous step */

7             $\Delta(\sigma^{-2}) = \sigma_{\text{new}}^{-2} - \sigma_{\text{prev}}^{-2}$

8             $\Delta\sigma^2 = \sigma_{\text{new}}^2 - \sigma_{\text{prev}}^2$

9             $\boldsymbol{\Sigma}_{0\,|\,t}^{-1} = \boldsymbol{\Sigma}_{0\,|\,t}^{-1} + \Delta(\sigma^{-2})\boldsymbol{I}$

10            $\boldsymbol{\Sigma}_{0\,|\,t} = (\boldsymbol{\Sigma}_{0\,|\,t}^{-1})^{-1}$

/* Transfer previous mu to new noise level */

11            $\boldsymbol{\mu}_{\text{transferred}} = \boldsymbol{x}_{\text{prev}} + \sigma_{\text{new}}^2(\sigma_{\text{next}}^2\boldsymbol{I} - \frac{\Delta\sigma^2}{\sigma_{\text{prev}}^2}\boldsymbol{\Sigma}_{0\,|\,t})^{-1}(\boldsymbol{\mu}_{\text{prev}} - \boldsymbol{x}_{\text{prev}})$

/* Space update */

12            $\Delta\boldsymbol{x} = \boldsymbol{x}_{\text{new}} - \boldsymbol{x}_{\text{prev}}$

13            $\Delta\boldsymbol{e} = \sigma_{\text{new}}^2(\boldsymbol{\mu}_{\text{new}} - \boldsymbol{\mu}_{\text{transferred}})$

14            $\gamma = \frac{1}{\Delta\boldsymbol{e}^\top\Delta\boldsymbol{x}}$

15            $\boldsymbol{\Sigma}_{0\,|\,t} = \boldsymbol{\Sigma}_{0\,|\,t} - \frac{\boldsymbol{\Sigma}_{0\,|\,t}\Delta\boldsymbol{x}\Delta\boldsymbol{x}^\top\boldsymbol{\Sigma}_{0\,|\,t}}{\Delta\boldsymbol{x}^\top\boldsymbol{\Sigma}_{0\,|\,t}\Delta\boldsymbol{x}} + \frac{\Delta\boldsymbol{e}\Delta\boldsymbol{e}^\top}{\Delta\boldsymbol{e}^\top\Delta\boldsymbol{x}}$

16         **end**

/* Calculate reconstruction guidance */

17         $\nabla_{\boldsymbol{x}} \log p(\boldsymbol{y}|\boldsymbol{x}_{\text{new}}) = (\boldsymbol{y} - \boldsymbol{A}\boldsymbol{\mu}_{\text{new}})^\top(\boldsymbol{A}\boldsymbol{\Sigma}_{0\,|\,t}\boldsymbol{A}^\top + \sigma_y^2\boldsymbol{I})^{-1}\boldsymbol{A}\nabla_{\boldsymbol{x}}\boldsymbol{\mu}_{\text{new}}$

/* Fall back to approximation if guidance too large */

18         **if** $\|\sigma_{new}^2\nabla_{\boldsymbol{x}} \log p(\boldsymbol{y}\,|\,\boldsymbol{x}_{new})\| > 1$ **then**

19            $\nabla_{\boldsymbol{x}} \log p(\boldsymbol{y}|\boldsymbol{x}_{\text{new}}) = (\boldsymbol{y} - \boldsymbol{A}\boldsymbol{\mu}_{\text{new}})^\top(\boldsymbol{A}\boldsymbol{\Sigma}_{0\,|\,t}\boldsymbol{A}^\top + \sigma_y^2\boldsymbol{I})^{-1}\boldsymbol{A}\frac{\boldsymbol{\Sigma}_{0\,|\,t}}{t_{\text{new}}^2}$

20         **end**

/* Update state variables */

21         $\boldsymbol{\mu}_{\text{prev}} = \boldsymbol{\mu}_{\text{new}}$

22         $\boldsymbol{x}_{\text{prev}} = \boldsymbol{x}_{\text{new}}$

23         $\sigma_{\text{prev}} = \sigma_{\text{new}}$

24     **return** $\nabla_{\boldsymbol{x}} \log p(\boldsymbol{y}|\boldsymbol{x}_{\text{new}})$

---

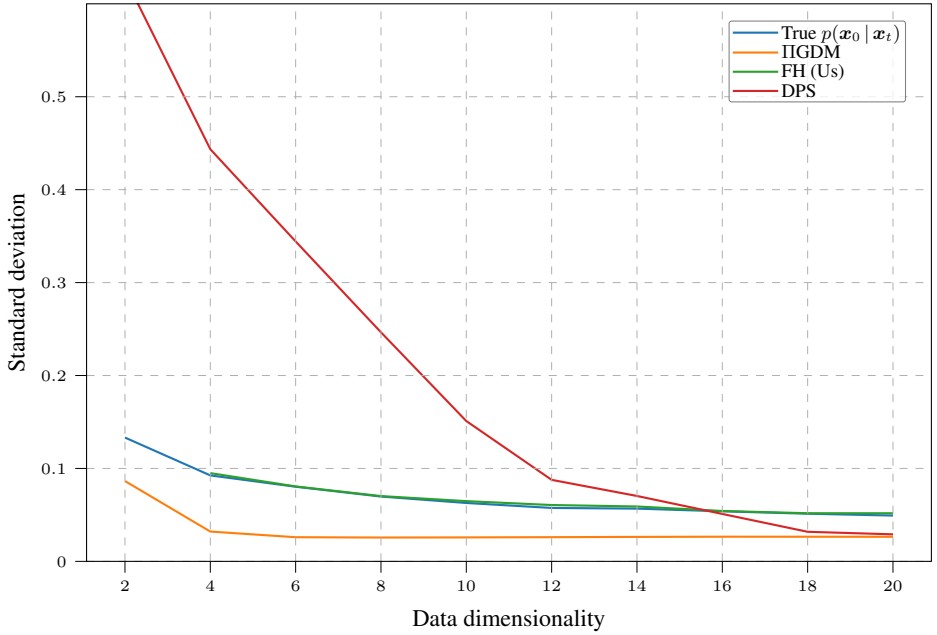

Figure 8: The standard deviation of posterior samples from different methods for the toy data discussed in App. E, showcasing the overconfidence problem caused by overestimated $\nabla_{\boldsymbol{x}_t} \log p(\boldsymbol{y} \mid \boldsymbol{x}_t)$.

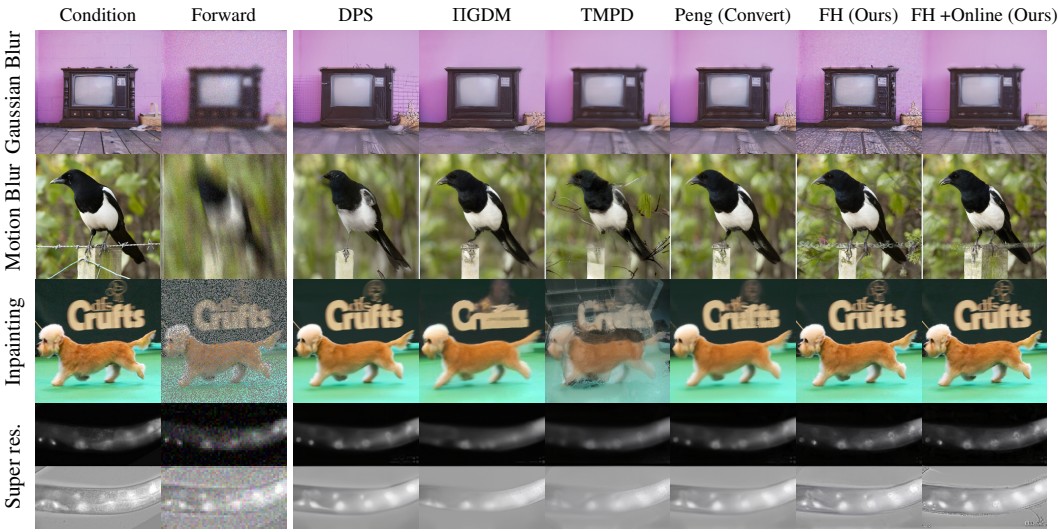

Figure 9: Qualitative results from 30-step Heun sampler.

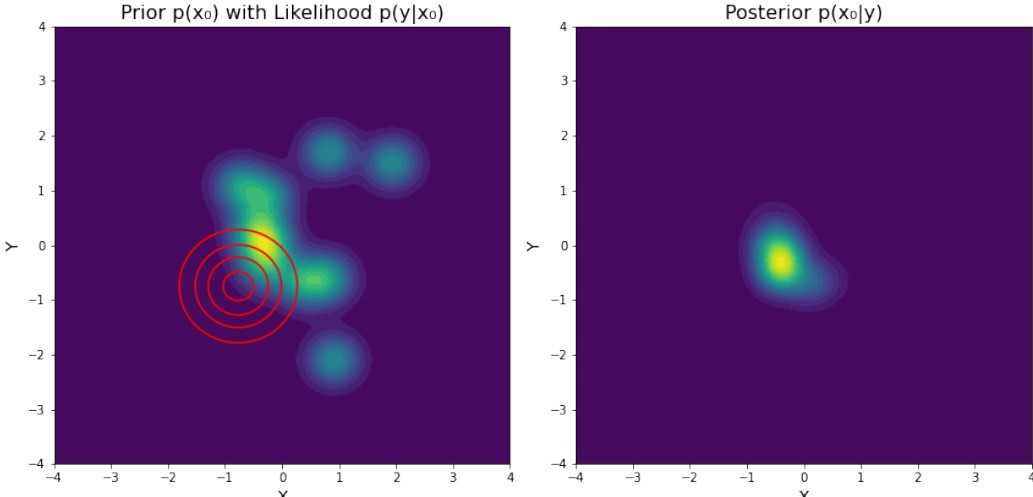

Figure 10: The posterior distribution of the toy data discussed in App. G and the prior distribution.

1. $\mathbb{C}\mathrm{ov}[\boldsymbol{x}_0 \,|\, \boldsymbol{x}_t] \approx \frac{\sigma(t)^2}{1+\sigma(t)^2}\boldsymbol{I}$, similarly to ΠGDM.

2. $\mathbb{C}\mathrm{ov}[\boldsymbol{x}_0 \,|\, \boldsymbol{x}_t]$ approximated with our method by initialising at the data covariance, but not performing space updates.

3. $\mathbb{C}\mathrm{ov}[\boldsymbol{x}_0 \,|\, \boldsymbol{x}_t]$ approximated with our method by initialising at the data covariance, and performing space updates.

4. $\mathbb{C}\mathrm{ov}[\boldsymbol{x}_0 \,|\, \boldsymbol{x}_t]$ approximated with our method and with space updates, but estimating the BFGS updates by calculating $\nabla_{\boldsymbol{x}_t} \log p(\boldsymbol{x}_t + \Delta\boldsymbol{x})$ and $\nabla_{\boldsymbol{x}_t} \log p(\boldsymbol{x}_t)$ explicitly, requiring 2 denoiser calls per step, but without error from the time updates affecting the BFGS updates. This method is also discussed in App. J.

For these experiments, we used the Euler–Maruyama sampler. The results are shown in Fig. 11. The ΠGDM covariance approximation is the furthest from the true value. Initialising at the data covariance helps, and adding the space updates decreases the error further. However, there is a clear gap in the standard method and using two score evaluations per step. We believe that this is due to the errors from the time updates affecting the BFGS updates.

We also perform an ablation comparing a deterministic Euler sampler and the stochastic Euler–Maryama sampler with the fourth method, and varying diffusion step count. The results are shown in Fig. 12. Whereas for the deterministic sampler, the covariance estimate does not significantly improve after a certain point, for the stochastic sampler, the error approaches zero across the sampling steps with more steps. This is because the reason the inherent curvature in the ODE path does not signifcantly change after increasing the step count above a certain limit. With a stochastic sampler, the generative path explores a slightly different direction at each step, and the BFSG updates get information from the curvature in all directions.

## H    ADDITIONAL RESULTS ON IMAGE MODELS

Here we list full additional results from:

1. ImageNet 256×256 results with Euler solver for 15, 30, 50, and 100 steps, in Table 2.

2. ImageNet 256×256 results with Heun solver for 15, 30, 50, and 100 steps, in Table 3.

For each task, we use 1000 samples from the ImageNet test set. We also evaluate DDNM+(Wang et al., 2023) and DiffPIR(Zhu et al., 2023) for a more thorough to different types of methods, although these can not directly be interpreted as reconstruction guidance with specific covariances. For DiffPIR, we use the implementation of (Peng et al., 2024), where they

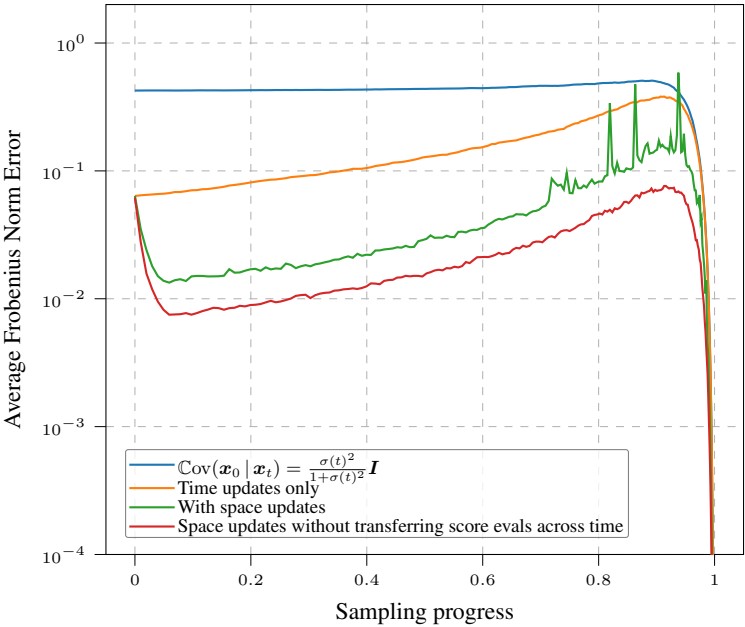

Figure 11: The Frobenius norm of the difference between the true covariance and the estimated covariance for different methods for different sampling steps (0 corresponds to the maximum noise level and 1 corresponds to zero noise level). As pointed out in App. J, using the time updates to transfer scores for use with the BFGS updates can cause inaccuracies with low-dimensional data, making the curve slightly rough, although still below the one with only time updates.

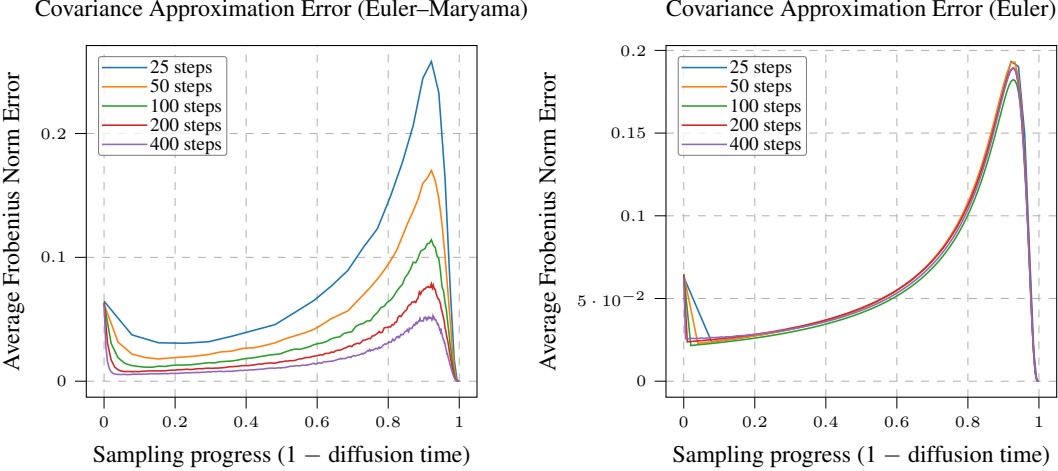

Figure 12: The Frobenius norm of the difference between the true covariance and the estimated covariance for different methods, with varying step count.

**Notes on hyperparameters** For DPS, Identity, Identity+Online updates, and DiffPIR, we tuned hyperparameters for each task using Gaussian blur as a baseline. We separately tuned the hyperparameters for the Euler and Heun solvers, and for each step count. While the optimal hyperparameters were similar for DiffPIR, Identity and Identity+Online updates, for DPS, the optimal values depended on the solver type and step count. We used 100 samples from the ImageNet validation set for tuning, and used these parameters for all experiments. The results are shown in Fig. 13,Fig. 14,Fig. 15 and Fig. 16.

## I MOTIVATIONS FOR THE DCT BASIS AND THE BFGS UPDATE

The reason that we chose the DCT over, e.g., the DFT basis is that it is purely real-valued, does not assume periodic boundaries, and in practice needs less coefficients to efficiently represent natural images. This is also one of the reasons for its use in the JPEG compression standard (Wallace, 1991). The BFGS update has the attractive property of preserving positive-semidefiniteness (as opposed to, e.g., the symmetric rank-1 update). This combines well with performing the updates in the denoiser covariance, which is positive-definite (as opposed to the Hessian). Compared to Davidon-Fletcher-Powell (DFP), the difference is that the BFGS update minimizes a weighted Frobenius norm for the size of the update in inverse covariance (Dennis & Moré, 1977), instead of the covariance directly. In the update formula in Eq. (23), if we use A=I and the obersation noise is low, the inverse term is simply the inverse covariance. Thus, it could stabilise the updates across iterations, but this is more speculative, and DFP could work in practice as well.

## J IMPLEMENTATION DETAILS

**The solver** We noticed that in large noise levels, it does not matter if the inversion in Eq. (23) is not exact, and we can set the tolerance quite high. We then defined a schedule for the tolerance such that it becomes lower towards the end. A lower tolerance towards the end of sampling is not an issue, since the covariance becomes closer to a diagonal, and the required matrix inverse becomes easier to calculate. In practice, we use the following schedule:

$$\sigma_{\text{max}} = 80$$
$$\sigma_{\text{min}} = 1$$
$$\text{rtol}_{\text{max}} = 1$$
$$\text{rtol}_{\text{min}} = 1e-14$$
$$p = 0.1$$
$$\sigma_{\text{clipped}} = \max(\min(\sigma, \sigma_{\text{max}}), \sigma_{\text{min}})$$
$$\text{log\_factor} = \left( \frac{\log_{10}(\sigma_{\text{clipped}}) - \log_{10}(\sigma_{\text{min}})}{\log_{10}(\sigma_{\text{max}}) - \log_{10}(\sigma_{\text{min}})} \right)^p \tag{72}$$
$$\text{log\_rtol} = \text{log\_factor} \cdot (\log_{10}(\text{rtol}_{\text{max}}) - \log_{10}(\text{rtol}_{\text{min}})) + \log_{10}(\text{rtol}_{\text{min}}) \tag{73}$$
$$\text{rtol} = 10^{\text{log\_rtol}}, \tag{74}$$

where $\text{rtol}_{\text{max}}$ and $\text{rtol}_{\text{min}}$ control the maximum and minimum relative tolerances of the solver, and $\sigma_{\text{max}}$ and $\sigma_{\text{min}}$ control the noise levels outside of which the tolerances are $\text{rtol}_{\text{max}}$ and $\text{rtol}_{\text{min}}$, respectively.

Note that scheduling the solver in this way does not improve image quality. Instead, it improves inference speed considerably, to the point where the solver is not a bottleneck anymore. Instead of using a standard off-the-shelf conjugate gradient implementation, we implemented one ourselves in PyTorch to utilize the speedup from the GPU.

Also for TMPD, we created a schedule for the conjugate gradient since a constant low tolerance slowed down the computation quite a bit. For TMPD, we use a standard scipy implementation that

Table 2: Results with the **Euler solver**. Our model performs especially well at small step sizes and remains competitive at larger step counts as well. DDNM+ is designed to enforce consistency with the measurement in cases where the measurement operator has a clearly defined nullspace, such as inpainting and super-resolution, potentially affecting the good PSNR and SSIM results there. In contrast, DDNM+ struggles with our Gaussian blur kernels. Motion blur results are not presented, as the code of DDNM+ assumes separable kernels.

| | Method | Deblur (Gaussian) | | | Inpainting (Random) | | | Deblur (Motion) | | | Super res. (4×) | | |
|---|---|---|---|---|---|---|---|---|---|---|---|---|---|
| | | PSNR↑ | SSIM↑ | LPIPS↓ | PSNR↑ | SSIM↑ | LPIPS↓ | PSNR↑ | SSIM↑ | LPIPS↓ | PSNR↑ | SSIM↑ | LPIPS↓ |
| **15 steps** | DPS | 19.94 | 0.444 | 0.572 | 20.68 | 0.494 | 0.574 | 17.02 | 0.354 | 0.646 | 19.85 | 0.460 | 0.590 |
| | ΠGDM | 20.29 | 0.474 | 0.574 | 19.87 | 0.468 | 0.598 | 19.21 | 0.429 | 0.602 | 20.17 | 0.474 | 0.582 |
| | TMPD | 22.56 | 0.572 | 0.486 | 17.70 | 0.447 | 0.589 | 20.40 | 0.481 | 0.567 | 21.15 | 0.517 | 0.541 |
| | Peng Convert | 22.53 | 0.563 | 0.490 | 22.23 | 0.579 | 0.489 | 20.46 | 0.475 | 0.556 | 21.92 | 0.541 | 0.517 |
| | Peng Analytic | 22.52 | 0.563 | 0.490 | 22.14 | 0.574 | 0.494 | 20.46 | 0.475 | 0.556 | 21.92 | 0.541 | 0.517 |
| | DDNM+ | 7.21 | 0.029 | 0.822 | 23.95 | 0.667 | 0.352 | – | – | – | **24.30** | **0.669** | 0.398 |
| | DiffPIR | 22.77 | 0.575 | 0.403 | 16.10 | 0.284 | 0.661 | 19.75 | 0.381 | 0.527 | 21.76 | 0.540 | 0.436 |
| | Identity | 22.91 | 0.594 | 0.384 | 18.83 | 0.397 | 0.590 | 20.06 | 0.393 | 0.506 | 22.65 | 0.589 | 0.412 |
| | Identity+online | 23.08 | 0.606 | 0.385 | 18.86 | 0.397 | 0.590 | 20.31 | 0.418 | 0.492 | 22.76 | 0.597 | 0.414 |
| | FH | 23.41 | 0.625 | **0.373** | 24.76 | 0.702 | 0.327 | 21.69 | 0.534 | 0.447 | 23.39 | 0.632 | **0.390** |
| | FH+online | **23.57** | **0.635** | 0.378 | **25.29** | **0.731** | **0.315** | 21.83 | 0.548 | 0.442 | 23.31 | 0.624 | 0.393 |
| **30 steps** | DPS | 21.76 | 0.527 | 0.463 | 24.84 | 0.678 | 0.386 | 18.22 | 0.389 | 0.582 | 23.00 | 0.593 | 0.440 |
| | ΠGDM | 22.27 | 0.559 | 0.468 | 21.24 | 0.518 | 0.517 | 21.15 | 0.508 | 0.503 | 22.11 | 0.552 | 0.479 |
| | TMPD | 22.92 | 0.591 | 0.451 | 18.27 | 0.465 | 0.563 | 20.71 | 0.495 | 0.538 | 21.59 | 0.536 | 0.507 |
| | Peng Convert | 23.61 | 0.627 | 0.405 | 23.74 | 0.648 | 0.403 | **21.99** | **0.553** | 0.463 | 23.21 | 0.608 | 0.430 |
| | Peng Analytic | 23.61 | 0.627 | 0.405 | 23.59 | 0.640 | 0.410 | **21.99** | 0.552 | 0.463 | 23.22 | 0.608 | 0.430 |
| | DDNM+ | 7.51 | 0.033 | 0.814 | **26.66** | **0.769** | 0.272 | – | – | – | **24.09** | **0.657** | 0.418 |
| | DiffPIR | 22.34 | 0.552 | 0.404 | 15.94 | 0.262 | 0.667 | 19.38 | 0.368 | 0.523 | 21.25 | 0.512 | 0.443 |
| | Identity | 23.15 | 0.602 | 0.374 | 18.75 | 0.402 | 0.578 | 20.14 | 0.406 | 0.494 | 22.82 | 0.588 | 0.405 |
| | Identity+online | 23.38 | 0.621 | 0.359 | 20.07 | 0.443 | 0.529 | 20.47 | 0.420 | 0.467 | 23.38 | 0.622 | 0.383 |
| | FH | 23.56 | 0.630 | 0.353 | 26.00 | 0.758 | **0.255** | 21.79 | 0.537 | 0.410 | 23.38 | 0.624 | **0.371** |
| | FH+online | **23.66** | **0.636** | 0.359 | 26.17 | 0.766 | 0.268 | 21.89 | 0.548 | 0.409 | 23.46 | 0.629 | 0.375 |
| **50 steps** | DPS | 22.66 | 0.579 | 0.411 | 26.31 | 0.761 | 0.297 | 19.05 | 0.428 | 0.537 | 23.79 | 0.642 | 0.375 |
| | ΠGDM | 22.64 | 0.577 | 0.434 | 21.67 | 0.536 | 0.484 | 21.56 | 0.526 | 0.468 | 22.48 | 0.571 | 0.442 |
| | TMPD | 23.09 | 0.600 | 0.434 | 18.50 | 0.472 | 0.551 | 20.83 | 0.500 | 0.524 | 21.76 | 0.543 | 0.492 |
| | Peng Convert | **23.81** | **0.638** | 0.377 | 24.75 | 0.698 | 0.346 | **22.30** | **0.567** | 0.430 | 23.44 | 0.622 | 0.400 |
| | Peng Analytic | **23.81** | **0.638** | 0.378 | 24.47 | 0.683 | 0.360 | **22.30** | **0.567** | 0.430 | 23.44 | 0.622 | 0.400 |
| | DDNM+ | 7.83 | 0.038 | 0.806 | 27.15 | 0.771 | 0.301 | – | – | – | **24.05** | **0.655** | 0.424 |
| | DiffPIR | 22.10 | 0.539 | 0.407 | 15.82 | 0.251 | 0.670 | 19.17 | 0.358 | 0.525 | 21.00 | 0.498 | 0.448 |
| | Identity | 23.18 | 0.602 | 0.360 | 19.64 | 0.436 | 0.536 | 19.92 | 0.373 | 0.497 | 23.11 | 0.600 | 0.385 |
| | Identity+online | 23.47 | 0.620 | 0.370 | 19.46 | 0.409 | 0.552 | 20.74 | 0.453 | 0.453 | 23.20 | 0.607 | 0.399 |
| | FH | 23.43 | 0.622 | **0.348** | 25.94 | 0.760 | **0.238** | 21.56 | 0.523 | 0.406 | 23.21 | 0.614 | **0.366** |
| | FH+online | 23.59 | 0.631 | 0.353 | 26.08 | **0.770** | 0.252 | 21.71 | 0.534 | 0.406 | 23.33 | 0.620 | 0.369 |
| **100 steps** | DPS | 23.36 | 0.615 | 0.379 | 26.61 | 0.800 | 0.229 | 20.05 | 0.473 | 0.492 | 23.36 | 0.622 | 0.366 |
| | ΠGDM | 22.76 | 0.585 | 0.408 | 21.97 | 0.551 | 0.459 | 21.71 | 0.533 | 0.440 | 22.63 | 0.580 | 0.416 |
| | TMPD | 23.18 | 0.604 | 0.423 | 18.67 | 0.477 | 0.543 | 20.90 | 0.502 | 0.514 | 21.88 | 0.548 | 0.480 |
| | Peng Convert | **23.74** | **0.638** | 0.358 | 24.89 | 0.706 | 0.332 | **22.36** | **0.570** | 0.407 | 23.46 | 0.625 | 0.379 |
| | Peng Analytic | 23.73 | 0.637 | 0.358 | 24.67 | 0.694 | 0.345 | **22.36** | **0.570** | 0.407 | 23.46 | 0.625 | 0.379 |
| | DDNM+ | 8.77 | 0.054 | 0.786 | 28.28 | **0.809** | 0.278 | – | – | – | 23.55 | 0.630 | 0.450 |
| | DiffPIR | 21.88 | 0.527 | 0.410 | 15.69 | 0.241 | 0.672 | 18.95 | 0.345 | 0.529 | 20.78 | 0.485 | 0.451 |
| | Identity | 23.11 | 0.596 | 0.361 | 19.66 | 0.439 | 0.528 | 19.72 | 0.358 | 0.501 | 23.04 | 0.594 | 0.384 |
| | Identity+online | 23.43 | 0.616 | 0.373 | 19.65 | 0.420 | 0.538 | 20.77 | 0.459 | 0.447 | 23.08 | 0.599 | 0.403 |
| | FH | 23.24 | 0.613 | **0.346** | 25.71 | 0.755 | 0.233 | 21.35 | 0.510 | **0.407** | 23.03 | 0.604 | **0.364** |
| | FH+online | 23.32 | 0.616 | 0.356 | 25.73 | 0.764 | 0.243 | 21.37 | 0.509 | 0.417 | 23.17 | 0.609 | 0.370 |

Table 3: Results with the **Heun solver**. Our model performs especially well at small step sizes and remains competitive at larger step counts as well. DDNM+ is designed to enforce consistency with the measurement in cases where the measurement operator has a clearly defined nullspace, such as inpainting and super-resolution, potentially affecting the good PSNR and SSIM results there. In contrast, DDNM+ struggles with our Gaussian blur kernels. Motion blur results are not presented, as the code of DDNM+ assumes separable kernels.

| | Method | Deblur (Gaussian) | | | Inpainting (Random) | | | Deblur (Motion) | | | Super res. (4×) | | |
|---|---|---|---|---|---|---|---|---|---|---|---|---|---|
| | | PSNR↑ | SSIM↑ | LPIPS↓ | PSNR↑ | SSIM↑ | LPIPS↓ | PSNR↑ | SSIM↑ | LPIPS↓ | PSNR↑ | SSIM↑ | LPIPS↓ |
| 15 steps | DPS | 19.94 | 0.444 | 0.572 | 20.68 | 0.494 | 0.574 | 17.02 | 0.354 | 0.646 | 19.85 | 0.460 | 0.590 |
| | ΠGDM | 20.30 | 0.475 | 0.574 | 19.87 | 0.468 | 0.598 | 19.21 | 0.429 | 0.602 | 20.17 | 0.474 | 0.582 |
| | TMPD | 23.08 | 0.597 | 0.420 | 18.99 | 0.481 | 0.539 | 20.80 | 0.491 | 0.514 | 21.88 | 0.545 | 0.476 |
| | Peng Convert | 22.53 | 0.563 | 0.490 | 22.23 | 0.579 | 0.489 | 20.46 | 0.475 | 0.556 | 21.92 | 0.541 | 0.517 |
| | Peng Analytic | 22.53 | 0.563 | 0.490 | 22.14 | 0.574 | 0.494 | 20.46 | 0.475 | 0.556 | 21.92 | 0.541 | 0.517 |
| | DDNM+ | 7.21 | 0.029 | 0.822 | 23.95 | 0.667 | 0.352 | – | – | – | **24.30** | **0.669** | 0.398 |
| | DiffPIR | 22.77 | 0.575 | 0.403 | 16.10 | 0.284 | 0.661 | 19.75 | 0.381 | 0.527 | 21.76 | 0.540 | 0.436 |
| | Identity | 22.91 | 0.594 | 0.384 | 18.83 | 0.397 | 0.590 | 20.06 | 0.393 | 0.506 | 22.65 | 0.589 | 0.412 |
| | Identity+online | 23.08 | 0.606 | 0.385 | 18.86 | 0.397 | 0.590 | 20.31 | 0.418 | 0.492 | 22.76 | 0.597 | 0.414 |
| | FH | 23.39 | 0.624 | **0.372** | 24.73 | 0.701 | 0.327 | 21.69 | 0.534 | 0.446 | 23.30 | 0.624 | **0.390** |
| | FH+online | **23.54** | **0.634** | 0.378 | **25.25** | **0.728** | **0.317** | **21.84** | **0.549** | **0.441** | 23.39 | 0.632 | 0.394 |
| 30 steps | DPS | 21.76 | 0.527 | 0.463 | 24.84 | 0.678 | 0.387 | 18.22 | 0.389 | 0.582 | 23.00 | 0.593 | 0.440 |
| | ΠGDM | 22.27 | 0.559 | 0.468 | 21.24 | 0.518 | 0.517 | 21.16 | 0.508 | 0.503 | 22.11 | 0.553 | 0.478 |
| | TMPD | 23.16 | 0.602 | 0.415 | 18.85 | 0.481 | 0.537 | 20.91 | 0.500 | 0.507 | 21.94 | 0.549 | 0.472 |
| | Peng Convert | 23.61 | 0.627 | 0.405 | 23.74 | 0.648 | 0.403 | **21.99** | **0.553** | 0.463 | 23.22 | 0.608 | 0.430 |
| | Peng Analytic | 23.61 | 0.626 | 0.405 | 23.59 | 0.640 | 0.411 | **21.99** | 0.552 | 0.463 | 23.21 | 0.608 | 0.430 |
| | DDNM+ | 7.51 | 0.033 | 0.814 | **26.66** | **0.769** | 0.272 | – | – | – | **24.09** | **0.657** | 0.418 |
| | DiffPIR | 22.34 | 0.552 | 0.404 | 15.94 | 0.262 | 0.667 | 19.38 | 0.368 | 0.523 | 21.25 | 0.512 | 0.443 |
| | Identity | 23.15 | 0.602 | 0.374 | 18.75 | 0.402 | 0.578 | 20.14 | 0.406 | 0.494 | 22.82 | 0.588 | 0.405 |
| | Identity+online | 23.38 | 0.621 | 0.359 | 20.07 | 0.443 | 0.529 | 20.47 | 0.420 | 0.467 | 23.38 | 0.622 | 0.383 |
| | FH | 23.55 | 0.630 | 0.353 | 26.00 | 0.757 | **0.256** | 21.80 | 0.538 | 0.411 | 23.38 | 0.623 | **0.372** |
| | FH+online | **23.62** | **0.635** | 0.358 | 26.18 | 0.767 | 0.268 | 21.88 | 0.547 | **0.410** | 23.44 | 0.628 | 0.375 |
| 50 steps | DPS | 22.66 | 0.578 | 0.412 | 26.31 | 0.761 | 0.296 | 19.05 | 0.428 | 0.537 | 23.80 | 0.642 | 0.375 |
| | ΠGDM | 22.64 | 0.577 | 0.435 | 21.67 | 0.536 | 0.484 | 21.56 | 0.526 | 0.468 | 22.48 | 0.571 | 0.442 |
| | TMPD | 23.20 | 0.605 | 0.414 | 18.83 | 0.482 | 0.536 | 20.93 | 0.502 | 0.504 | 21.96 | 0.550 | 0.471 |
| | Peng Convert | 23.81 | **0.638** | 0.377 | 24.75 | 0.698 | 0.346 | **22.30** | **0.567** | 0.429 | 23.44 | 0.622 | 0.400 |
| | Peng Analytic | 23.80 | **0.638** | 0.378 | 24.47 | 0.683 | 0.360 | **22.30** | **0.567** | 0.430 | 23.44 | 0.622 | 0.400 |
| | DDNM+ | 7.83 | 0.038 | 0.806 | **27.15** | **0.771** | 0.301 | – | – | – | **24.05** | **0.655** | 0.424 |
| | DiffPIR | 22.10 | 0.539 | 0.407 | 15.82 | 0.251 | 0.670 | 19.17 | 0.358 | 0.525 | 21.00 | 0.498 | 0.448 |
| | Identity | 23.18 | 0.602 | 0.360 | 19.64 | 0.436 | 0.536 | 19.92 | 0.373 | 0.497 | 23.11 | 0.600 | 0.385 |
| | Identity+online | 23.47 | 0.620 | 0.370 | 19.46 | 0.409 | 0.552 | 20.74 | 0.453 | 0.453 | 23.20 | 0.607 | 0.399 |
| | FH | 23.44 | 0.623 | **0.348** | 25.95 | 0.760 | **0.237** | 21.58 | 0.523 | 0.406 | 23.22 | 0.614 | **0.367** |
| | FH+online | 23.60 | 0.631 | 0.353 | 26.10 | **0.772** | 0.250 | 21.73 | 0.535 | 0.406 | 23.31 | 0.619 | 0.369 |
| 100 steps | DPS | 23.36 | 0.615 | 0.378 | 26.61 | 0.800 | 0.228 | 20.05 | 0.473 | 0.492 | 23.36 | 0.622 | 0.366 |
| | ΠGDM | 22.76 | 0.585 | 0.408 | 21.97 | 0.551 | 0.459 | 21.71 | 0.533 | 0.440 | 22.63 | 0.580 | 0.416 |
| | TMPD | 23.22 | 0.606 | 0.412 | 18.83 | 0.482 | 0.536 | 20.95 | 0.502 | 0.504 | 21.98 | 0.551 | 0.469 |
| | Peng Convert | **23.74** | **0.638** | 0.358 | 24.89 | 0.706 | 0.332 | **22.36** | **0.570** | 0.407 | 23.46 | 0.625 | 0.379 |
| | Peng Analytic | 23.73 | 0.637 | 0.358 | 24.67 | 0.694 | 0.345 | **22.36** | **0.570** | 0.407 | 23.46 | 0.625 | 0.379 |
| | DDNM+ | 8.77 | 0.054 | 0.786 | 28.28 | **0.809** | 0.278 | – | – | – | 23.55 | 0.630 | 0.450 |
| | DiffPIR | 21.88 | 0.527 | 0.410 | 15.69 | 0.241 | 0.672 | 18.95 | 0.345 | 0.529 | 20.78 | 0.485 | 0.451 |
| | Identity | 23.11 | 0.596 | 0.361 | 19.66 | 0.439 | 0.528 | 19.72 | 0.358 | 0.501 | 23.04 | 0.594 | 0.384 |
| | Identity+online | 23.43 | 0.616 | 0.373 | 19.65 | 0.420 | 0.538 | 20.77 | 0.458 | 0.447 | 23.08 | 0.599 | 0.403 |
| | FH | 23.25 | 0.613 | **0.346** | 25.71 | 0.755 | 0.233 | 21.35 | 0.509 | 0.408 | 23.02 | 0.604 | **0.364** |
| | FH+online | 23.35 | 0.616 | 0.357 | 25.75 | 0.765 | 0.242 | 21.40 | 0.512 | 0.414 | 23.15 | 0.608 | 0.371 |

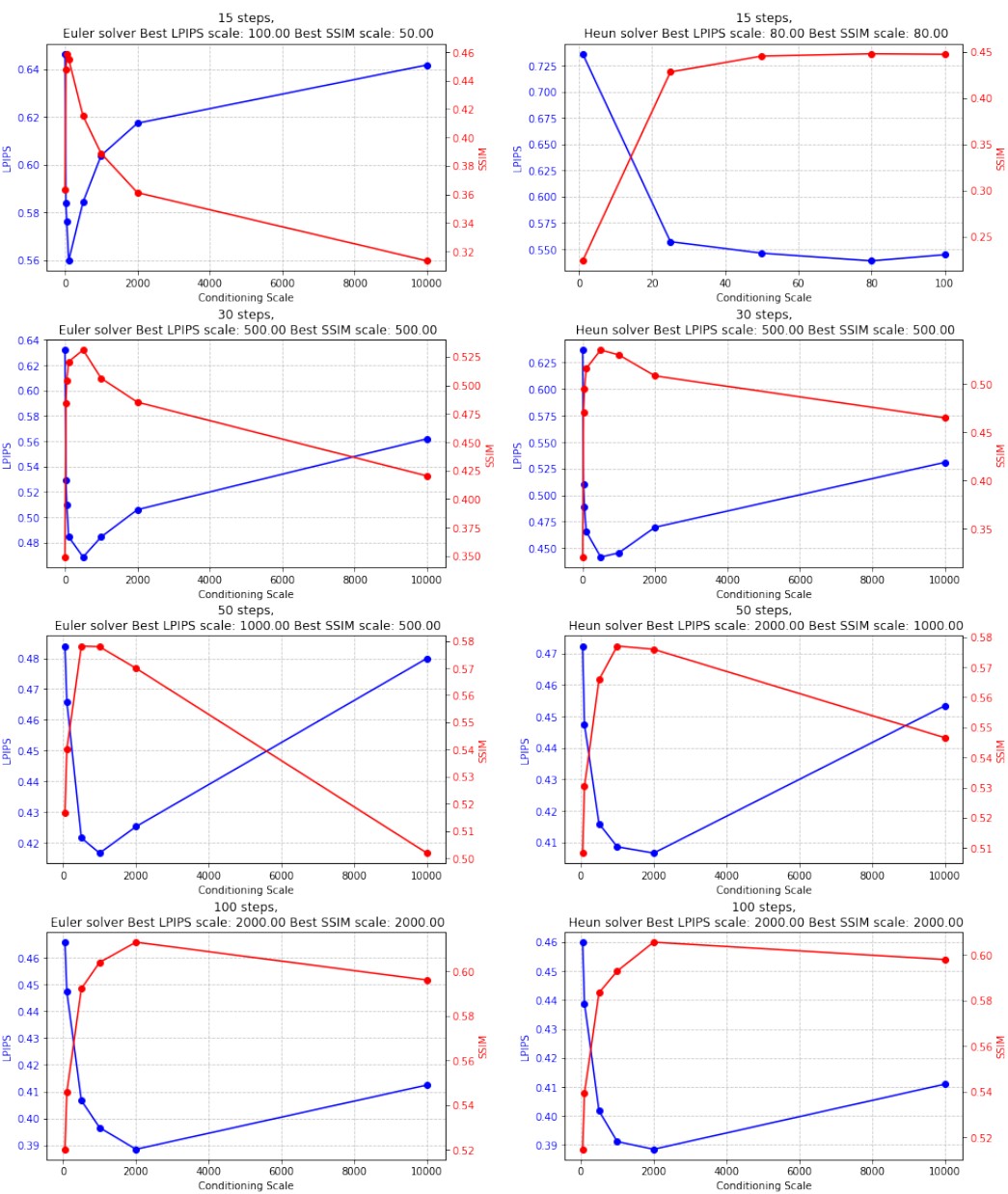

Figure 13: LPIPS and SSIM metrics across different solver steps and conditioning scales for DPS. The optimal LPIPS values are used in the experiments.

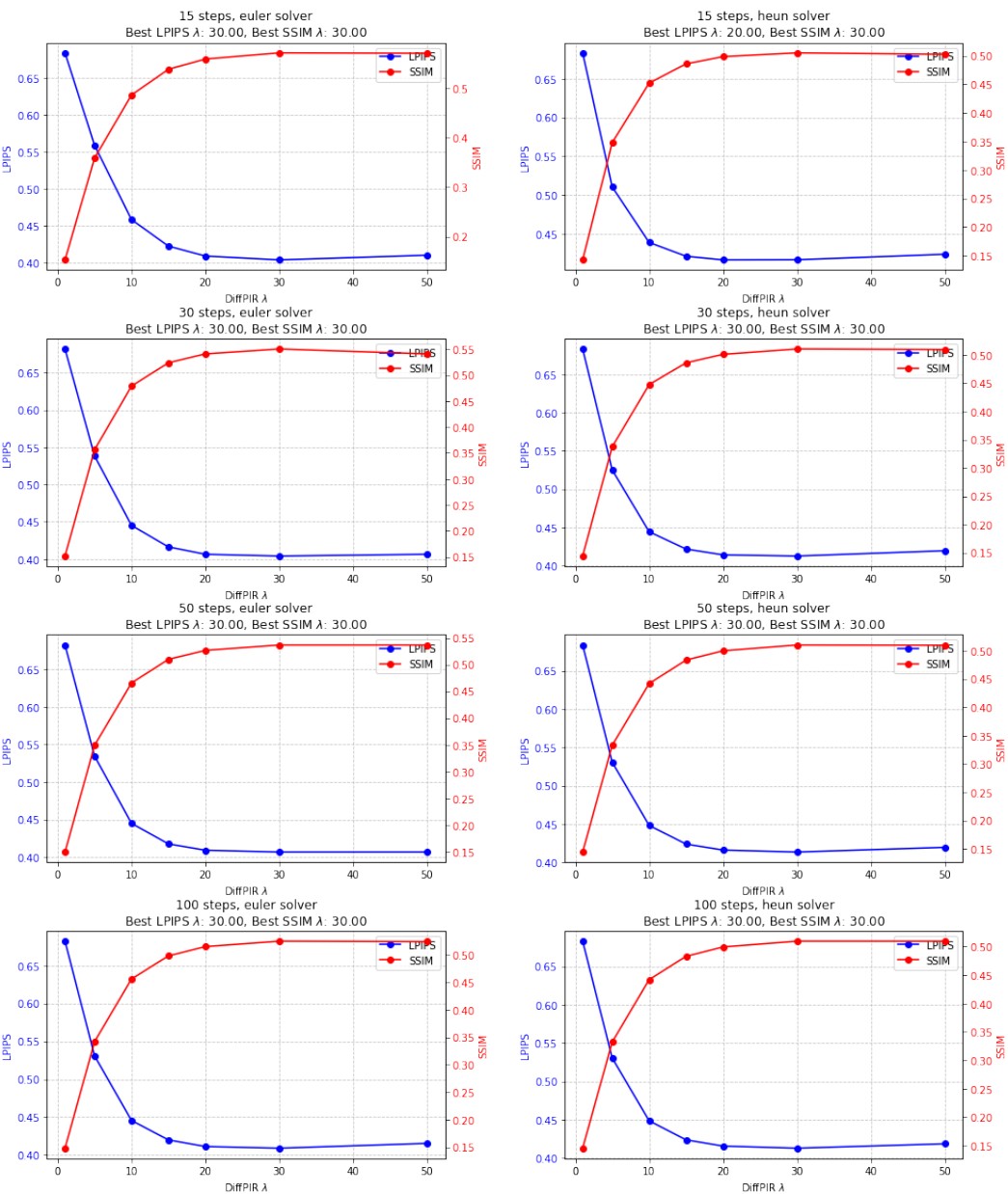

Figure 14: LPIPS and SSIM metrics across different solver steps and $\lambda$ values for DiffPIR. The optimal LPIPS values are used in the experiments.

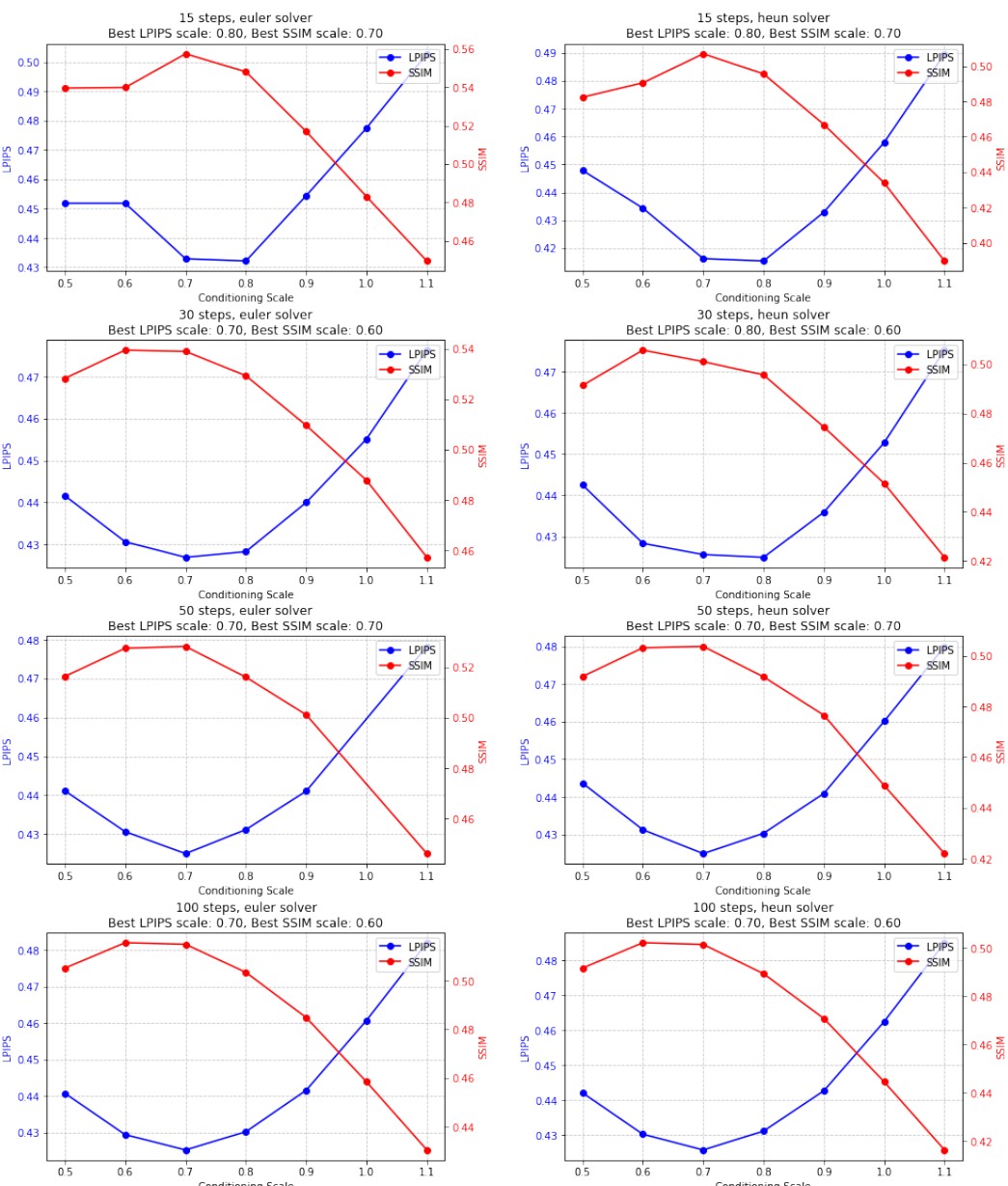

Figure 15: LPIPS and SSIM metrics across different solver steps and conditioning scales for our method with the identity base covariance. The optimal LPIPS values are used in the experiments.

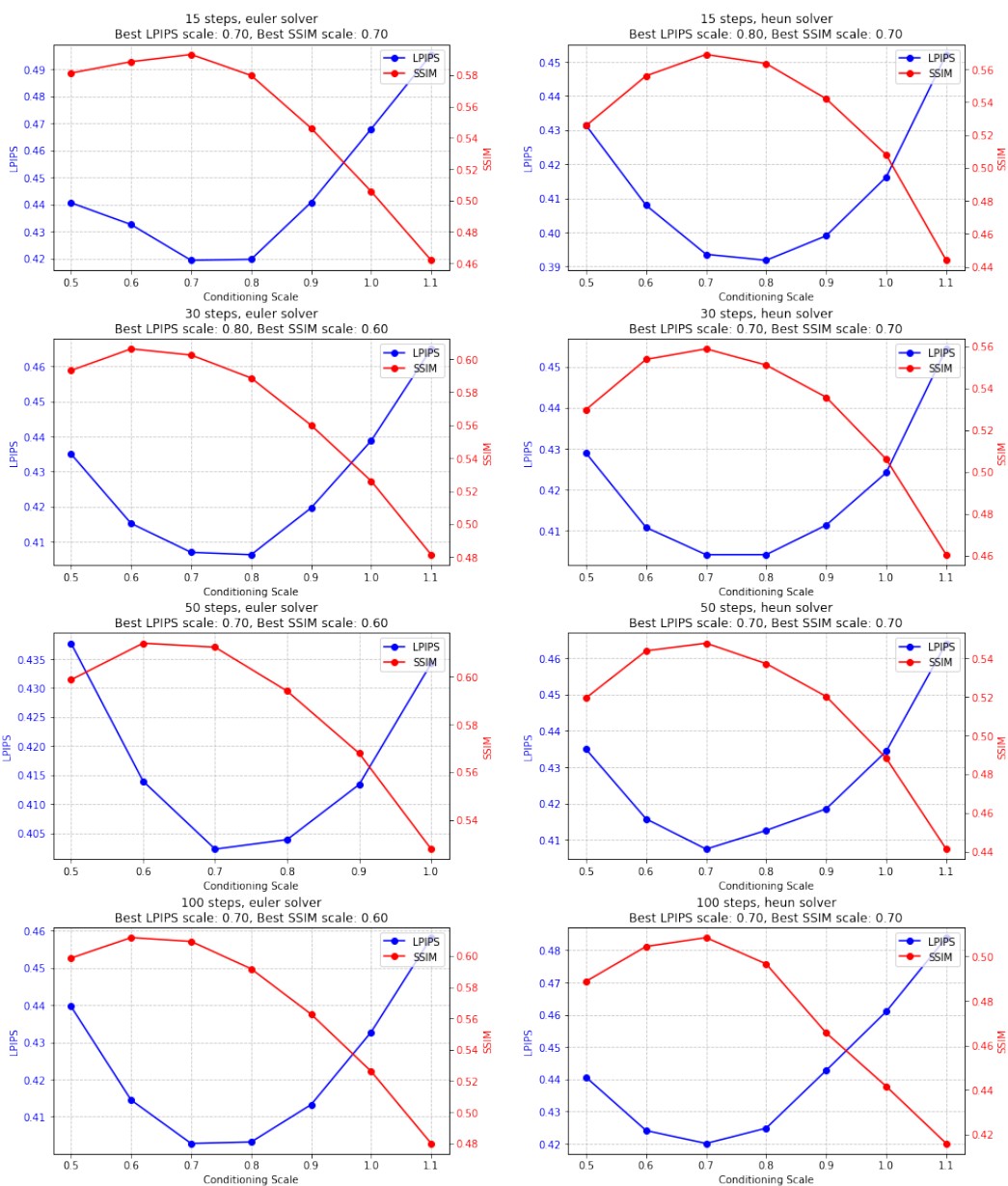

Figure 16: LPIPS and SSIM metrics across different solver steps and conditioning scales for our method with the identity base covariance and online updates. The optimal LPIPS values are used in the experiments.

is parameterized that is parameterized in terms of

$$\sigma_{\text{max}} = 80$$
$$\sigma_{\text{min}} = 1$$
$$\text{tol}_{\text{max}} = 1$$
$$\text{tol}_{\text{min}} = 1e - 14$$
$$p = 0.05$$
$$\sigma_{\text{clipped}} = \max(\min(\sigma, \sigma_{\text{max}}), \sigma_{\text{min}})$$

$$\text{log\_factor} = \left( \frac{\log_{10}(\sigma_{\text{clipped}}) - \log_{10}(\sigma_{\text{min}})}{\log_{10}(\sigma_{\text{max}}) - \log_{10}(\sigma_{\text{min}})} \right)^p \tag{75}$$

$$\text{log\_rtol} = \text{log\_factor} \cdot (\log_{10}(\text{rtol}_{\text{max}}) - \log_{10}(\text{rtol}_{\text{min}})) + \log_{10}(\text{rtol}_{\text{min}}) \tag{76}$$

$$\text{rtol} = 10^{\text{log\_rtol}} \tag{77}$$

We tried to choose the schedule such that this does not degrade the performance noticeably, but also allows us to run experiments in reasonable time.

**Range for the BFGS updates**   In practice, we do the space/BFGS updates for a range of $\sigma(t)$ values for image data, in particular $1 \leq \sigma(t) \leq 5$. A motivation for this choice is that we noticed the finite differences to not be numerically accurate at high noise levels, where the time updates $\Delta t$ and the space updates $\Delta \boldsymbol{x}$ are large. For low noise levels, it is also unnecessary, given that the covariance approaches $\sigma(t)^2 I$ in any case. How to apply the space updates in the optimal way is an interesting direction for future research.

**The solver**   We noticed that in large noise levels, it does not matter if the inversion in Eq. (23) is not exact, and we can set the tolerance quite high. We then defined a schedule for the tolerance such that it becomes lower towards the end. A lower tolerance towards the end of sampling is not an issue, since the covariance becomes closer to a diagonal, and the required matrix inverse becomes easier to calculate.

**Details on the low-dimensional experiments**   In the Gaussian mixture experiments, we did not apply the time updates for $\boldsymbol{\mu}_t(\boldsymbol{x})$, but instead evaluate $\boldsymbol{\mu}_{0 \,|\, t+\Delta t}(\boldsymbol{x})$ explicitly before applying the BFGS update. The reason is that on low-dimensional data, some of the prior samples are close to the actual data distribution. In that region, the time evolution is complex enough from the start that the denoiser mean time update coupled with the BFGS update in the next step sometimes causes numerical instability. To avoid additional score function evaluations, we could devise a schedule for when to apply the space updates.

**Measurement operators.**   We obtained the measurement operator definitions from (Peng et al., 2024), which in turn are based on the operators in (Chung et al., 2023). We use a noise level $\sigma_y = 0.1$ for all measurement models (data scaled to [-1,1]).

## K   ADDITIONAL RESULTS ON OPTIMAL GUIDANCE STRENGTH

Figure 17 shows the PSNR, SSIM and LPIPS scores for the Gaussian deblurring task on ImageNet 256×256 with different post-hoc guidance scales on the initially calculated guidance term $\nabla_{\boldsymbol{x}_t} \log p(\boldsymbol{y} \,|\, \boldsymbol{x}_t)$.

## L   COMPUTATIONAL REQUIREMENTS

The sweep to obtain the results in Table 1 was done with multiple NVIDIA V100 GPUs in a few hours, and can be obtained with a single V100s in less than a day of compute. For some of the methods, the matrix inversion in Eq. (23) can slow down generation considerably, although this is not as significant an overhead with our tolerance-optimized conjugate gradient implementation.

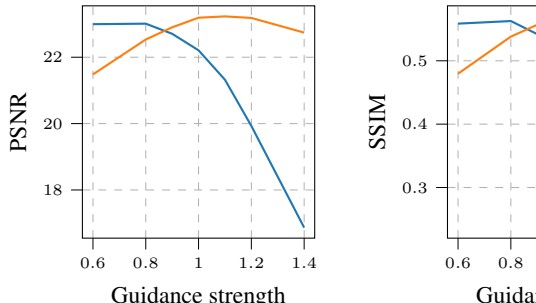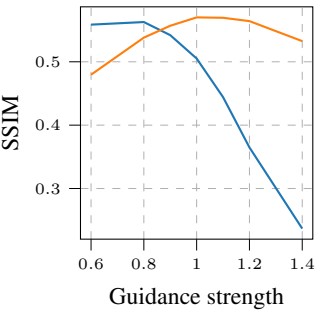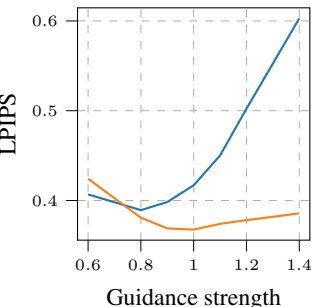

Figure 17: Different metrics with respect to guidance strength for 100 images from the Imagenet validation set, using an identity base covariance (blue) and a DCT-diagonal covariance (orange). With a better covariance approximation, the usefulness of adjusting the resulting guidance with post-hoc tricks becomes smaller.

# M  OPTIMAL $\nabla_{\boldsymbol{x}_t} \log p(\boldsymbol{x}_t \,|\, \boldsymbol{y})$ FOR GAUSSIAN MIXTURE DATA AND GAUSSIAN OBSERVATION

In this section, we derive the optimal gradient $\nabla_{\boldsymbol{x}_t} \log p(\boldsymbol{x}_t \,|\, \boldsymbol{y})$ for a situation with Gaussian mixture data and a Gaussian observation model. This is useful for performing toy experiments without having to retrain the model. Note that we can get the unconditional score from the end result by setting the observation noise $\Sigma_y$ to infinity.

**Model Definition**  We begin with the following components:

1. **Prior Distribution:** The prior on $\boldsymbol{x}_0$ is a Gaussian mixture model:

$$p(\boldsymbol{x}_0) = \sum_i w_i \mathcal{N}(\boldsymbol{x}_0 \,|\, \boldsymbol{\mu}_i, \boldsymbol{\Sigma}_i), \tag{78}$$

   where $w_i$ are the mixture weights, $\boldsymbol{\mu}_i$ are the mean vectors, and $Sigma_i$ are the covariance matrices for each mixture component.

2. **Likelihood:** The observation model is Gaussian:

$$p(\boldsymbol{y} \,|\, \boldsymbol{x}_0) = \mathcal{N}(\boldsymbol{y} \,|\, \boldsymbol{x}_0, \boldsymbol{\Sigma}_y) \tag{79}$$

   where $\boldsymbol{\Sigma}_y$ is the observation noise covariance.

3. **Transition Model:** The transition from $\boldsymbol{x}_0$ to $\boldsymbol{x}_t$ is modeled as:

$$p(\boldsymbol{x}_t \,|\, \boldsymbol{x}_0) = \mathcal{N}(\boldsymbol{x}_t \,|\, \boldsymbol{x}_0, \sigma(t)^2 \boldsymbol{I}) \tag{80}$$

   where $\sigma^2 \boldsymbol{I}$ is isotropic Gaussian noise with variance $\sigma(t)^2$.

## M.1  POSTERIOR DISTRIBUTION

Given these components, the posterior distribution $p(\boldsymbol{x}_0 \,|\, \boldsymbol{x}_t, \boldsymbol{y})$ is also a Gaussian mixture:

$$p(\boldsymbol{x}_0 \,|\, \boldsymbol{x}_t, \boldsymbol{y}) = \sum_i w_i' \mathcal{N}(\boldsymbol{x}_0 \,|\, \boldsymbol{\mu}_i', \, \Sigma_i'), \tag{81}$$

where

$$\Sigma_i'^{-1} = (\sigma(t)^2 \boldsymbol{I})^{-1} + \boldsymbol{\Sigma}_y^{-1} + \boldsymbol{\Sigma}_i^{-1} \tag{82}$$

$$\boldsymbol{\mu}_i' = \Sigma_i'((\sigma(t)^2 \boldsymbol{I})^{-1} \boldsymbol{x}_t + \boldsymbol{\Sigma}_y^{-1} \boldsymbol{y} + \boldsymbol{\Sigma}_i^{-1} \boldsymbol{\mu}_i) \tag{83}$$

$$w_i' \propto w_i \mathcal{N}(\boldsymbol{x}_t \,|\, \boldsymbol{\mu}_i, \sigma(t)^2 \boldsymbol{I} + \boldsymbol{\Sigma}_i) \mathcal{N}(\boldsymbol{y} \,|\, \boldsymbol{\mu}_i, \boldsymbol{\Sigma}_y + \boldsymbol{\Sigma}_i). \tag{84}$$

## M.2  CONDITIONAL EXPECTATION

The conditional expectation of $\boldsymbol{x}_0$ given $\boldsymbol{x}_t$ and $\boldsymbol{y}$ is:

$$\mathbb{E}[\boldsymbol{x}_0 \,|\, \boldsymbol{x}_t, \boldsymbol{y}] = \sum_i w_i' \boldsymbol{\mu}_i' \tag{85}$$

## M.3 DERIVATION OF THE GRADIENT

Now, let's derive the gradient $\nabla_{\boldsymbol{x}_t} \log p(\boldsymbol{x}_t \mid \boldsymbol{y})$:

1. We start with:

$$\nabla_{\boldsymbol{x}_t} \log p(\boldsymbol{x}_t \mid \boldsymbol{y}) = \nabla_{\boldsymbol{x}_t} \log \int p(\boldsymbol{x}_t \mid \boldsymbol{x}_0) p(\boldsymbol{x}_0 \mid \boldsymbol{y}) \, d\boldsymbol{x}_0 \tag{86}$$

2. Applying the chain rule and moving the gradient inside the integral:

$$\nabla_{\boldsymbol{x}_t} \log p(\boldsymbol{x}_t \mid \boldsymbol{y}) = \frac{\int \nabla_{\boldsymbol{x}_t} p(\boldsymbol{x}_t \mid \boldsymbol{x}_0) p(\boldsymbol{x}_0 \mid \boldsymbol{y}) d\boldsymbol{x}_0}{\int p(\boldsymbol{x}_t \mid \boldsymbol{x}_0) p(\boldsymbol{x}_0 \mid \boldsymbol{y}) d\boldsymbol{x}_0} \tag{87}$$

3. Given $p(\boldsymbol{x}_t \mid \boldsymbol{x}_0) = \mathcal{N}(\boldsymbol{x}_t \mid \boldsymbol{x}_0, \sigma(t)^2 \boldsymbol{I})$, we have:

$$\nabla_{\boldsymbol{x}_t} p(\boldsymbol{x}_t \mid \boldsymbol{x}_0) = -\frac{1}{\sigma(t)^2} (\boldsymbol{x}_t - \boldsymbol{x}_0) p(\boldsymbol{x}_t \mid \boldsymbol{x}_0) \tag{88}$$

4. Substituting this back:

$$\nabla_{\boldsymbol{x}_t} \log p(\boldsymbol{x}_t \mid \boldsymbol{y}) = -\frac{1}{\sigma(t)^2} \frac{\int (\boldsymbol{x}_t - \boldsymbol{x}_0) p(\boldsymbol{x}_t \mid \boldsymbol{x}_0) p(\boldsymbol{x}_0 \mid \boldsymbol{y}) \, d\boldsymbol{x}_0}{\int p(\boldsymbol{x}_t \mid \boldsymbol{x}_0) p(\boldsymbol{x}_0 \mid \boldsymbol{y}) \, d\boldsymbol{x}_0} \tag{89}$$

$$= -\frac{1}{\sigma(t)^2} (\boldsymbol{x}_t - \frac{\int \boldsymbol{x}_0 p(\boldsymbol{x}_t \mid \boldsymbol{x}_0) p(\boldsymbol{x}_0 \mid \boldsymbol{y}) \, d\boldsymbol{x}_0}{\int p(\boldsymbol{x}_t \mid \boldsymbol{x}_0) p(\boldsymbol{x}_0 \mid \boldsymbol{y}) \, d\boldsymbol{x}_0}) \tag{90}$$

$$= -\frac{1}{\sigma(t)^2} (\boldsymbol{x}_t - \mathbb{E}[\boldsymbol{x}_0 \mid \boldsymbol{x}_t, \boldsymbol{y}]) \tag{91}$$

## M.4 FINAL FORM OF THE GRADIENT

Therefore, the final form of the gradient is:

$$\nabla_{\boldsymbol{x}_t} \log p(\boldsymbol{x}_t \mid \boldsymbol{y}) = -\frac{1}{\sigma^2} (\boldsymbol{x}_t - \mathbb{E}[\boldsymbol{x}_0 \mid \boldsymbol{x}_t, \boldsymbol{y}]) \tag{92}$$

## M.5 DETAILED FORMULA FOR $\mathbb{E}[\boldsymbol{x}_0 \mid \boldsymbol{x}_t, \boldsymbol{y}]$

Let's expand the formula for $\mathbb{E}[\boldsymbol{x}_0 \mid \boldsymbol{x}_t, \boldsymbol{y}]$:

1. We start with the posterior distribution:

$$p(\boldsymbol{x}_0 \mid \boldsymbol{x}_t, \boldsymbol{y}) = \sum_i w_i' \mathcal{N}(\boldsymbol{x}_0 \mid \boldsymbol{\mu}_i', \boldsymbol{\Sigma}_i') \tag{93}$$

2. The expectation of this mixture is the weighted sum of the means:

$$\mathbb{E}[\boldsymbol{x}_0 \mid \boldsymbol{x}_t, \boldsymbol{y}] = \sum_i w_i' \boldsymbol{\mu}_i' \tag{94}$$

3. Expanding $\boldsymbol{\mu}_i'$:

$$\boldsymbol{\mu}_i' = \boldsymbol{\Sigma}_i' ((\sigma(t)^2 \boldsymbol{I})^{-1} \boldsymbol{x}_t + \boldsymbol{\Sigma}_y^{-1} \boldsymbol{y} + \boldsymbol{\Sigma}_i^{-1} \boldsymbol{\mu}_i) \tag{95}$$

4. Substituting this into the expectation formula:

$$\mathbb{E}[\boldsymbol{x}_0 \mid \boldsymbol{x}_t, \boldsymbol{y}] = \sum_i w_i' \boldsymbol{\Sigma}_i' ((\sigma(t)^2 \boldsymbol{I})^{-1} \boldsymbol{x}_t + \boldsymbol{\Sigma}_y^{-1} \boldsymbol{y} + \boldsymbol{\Sigma}_i^{-1} \boldsymbol{\mu}_i) \tag{96}$$

5. Rearranging:

$$\mathbb{E}[\boldsymbol{x}_0 \mid \boldsymbol{x}_t, \boldsymbol{y}] = (\sum_i w_i' \boldsymbol{\Sigma}_i' (\sigma(t)^2 \boldsymbol{I})^{-1}) \boldsymbol{x}_t + (\sum_i w_i' \boldsymbol{\Sigma}_i' \boldsymbol{\Sigma}_y^{-1}) \boldsymbol{y} + \sum_i w_i' \Sigma_i' \Sigma_i^{-1} \boldsymbol{\mu}_i \tag{97}$$

Let's define:

$$\boldsymbol{A} = \sum_i w_i' \boldsymbol{\Sigma}_i' (\sigma(t)^2 \boldsymbol{I})^{-1}, \tag{98}$$

$$\boldsymbol{B} = \sum_i w_i' \boldsymbol{\Sigma}_i' \boldsymbol{\Sigma}_y^{-1}, \tag{99}$$

$$\boldsymbol{c} = \sum_i w_i' \boldsymbol{\Sigma}_i' \boldsymbol{\Sigma}_i^{-1} \boldsymbol{\mu}_i. \tag{100}$$

Then we can write the final formula as:

$$\mathbb{E}[\boldsymbol{x}_0 \,|\, \boldsymbol{x}_t, \boldsymbol{y}] = \boldsymbol{A}\boldsymbol{x}_t + \boldsymbol{B}\boldsymbol{y} + \boldsymbol{c} \tag{101}$$

where:

$$w_i' \propto w_i \mathcal{N}(\boldsymbol{x}_t \,|\, \boldsymbol{\mu}_i, \sigma(t)^2 \boldsymbol{I} + \Sigma_i) \mathcal{N}(\boldsymbol{y} \,|\, \boldsymbol{\mu}_i, \boldsymbol{\Sigma}_y + \boldsymbol{\Sigma}_i), \tag{102}$$

$$\boldsymbol{\Sigma}_i' = ((\sigma(t)^2 \boldsymbol{I})^{-1} + \boldsymbol{\Sigma}_y^{-1} + \boldsymbol{\Sigma}_i^{-1})^{-1}. \tag{103}$$

This formula shows that $\mathbb{E}[\boldsymbol{x}_0 \,|\, \boldsymbol{x}_t, \boldsymbol{y}]$ is a linear combination of $\boldsymbol{x}_t$ and $\boldsymbol{y}$, plus a constant term. The matrices $A$ and $B$ determine how much the expectation depends on $\boldsymbol{x}_t$ and $\boldsymbol{y}$ respectively, while $c$ represents a constant offset based on the prior distribution.

