# OpenReview forum: "Free Hunch: Denoiser Covariance Estimation for Diffusion Models Without Extra Costs"
_ICLR.cc/2025/Conference — ICLR 2025 Poster_

### Official Review · Reviewer_d5jC · 2024-10-21

**Soundness:** 3
**Presentation:** 1
**Contribution:** 2
**Rating:** 6
**Confidence:** 3

**Summary:**

This paper introduce a L-BFGS-like method to maintain the covariance information of denoiser along the inference path to better solve the linear inverse problems via diffusion models.

**Strengths:**

1. It is novel to leverage L-BFGS way to approximate the covariance in a low-dim manner in DMs
2. the mathematical derivation is rigorous, though it is quite similar to the math of BFGS optimization
3. The design of initial covariance to be the data covariance is reasonable.

**Weaknesses:**

1. The presentation of the paper could be improved for clarity. The term 'conditional generation' is often associated with 'text-guided' or 'label-guided' generation. However, in this paper, it refers to conditions based on partially corrupted samples. While this interpretation is not incorrect and indeed falls under the classifier-guided framework, it might be confusing for readers at first glance.

2. There are typographical errors present in the paper, for instance, in equation (98). These need to be corrected for accuracy and better understanding.

3. The paper lacks a theoretical bound on the approximation error of the covariance using the proposed method. Both Equation 6 and the low-rank approximation involve approximation errors.

**Questions:**

see Weaknesses.

---

> ### Author Response · Authors · 2024-11-20
> **Answer to reviewer d5jC**
>
> We thank the reviewer for the useful feedback. We address the points that were raised below.
>
> **Presentation and clarity.** We agree that a more precise wording at the outset of the paper is useful to avoid confusion. In the abstract, we now refer to 'training-free guided generation' instead of the more generic 'conditional generation'. This class of methods is, in principle, applicable to other types of conditions, like text-conditional generation [1], although we focus on the linear inverse problem setting.
>
> **Typographical errors.** Thank you for raising our attention to these. We have gone through the text to make sure the typography is correct and removed typos and inconsistencies in notation.
>
> **Theoretical bound.** While a complete theoretical error analysis is beyond our current scope, the question prompted us to add several important insights to the paper: In Appendix G, we analyse the convergence of the error in covariance estimation with increasing step count, showing that stochastic solvers indeed converge towards the true covariance. In contrast, for an ODE solver the curvature of the ODE sets an upper bound on the amount of covariance information attainable. We also show that there is a clear progression of increasing accuracy going from:
> 1. An identity covariance $Cov[x_0|x_t]=\frac{\sigma(t)^2}{1+\sigma(t)^2}I$
> 2. Our method, but using only the time updates
> 3. Our method with time+space updates.
>
> Another potential theoretical approach to analyse the method would be to focus on the simplified case of no space updates and derive bounds on the errors introduced. This could be an interesting direction for future research.
>
> We hope that this answers the concerns of the reviewer, and are open to any further questions and discussion.
>
> References:
> [1] Song et al. "Loss-guided diffusion models for plug-and-play controllable generation."

---

> > ### Author Response · Authors · 2024-11-27
> >
> > Dear reviewer d5jC,
> >
> > We would like to thank again for your time in reviewing our paper, and providing constructive criticism. After the rebuttal, we believe that the presentation, experiments section and analysis of the method are clearly improved. Would you have the time to indicate whether we have addressed your concerns (to raise the score)? We are also happy to discuss more.

---

> > > ### Comment · Reviewer_d5jC · 2024-11-27
> > >
> > > Thanks for your reply, I would raise my score to 6 due to the additional experiments and theoretical analysis.
> > >
> > > However, I am not confident in whether the linear reverse problem comparison is valid or not. I am not an expert on these experiments. If other reviewers agree with your experiments, I would like to back up your acceptance due to the novelty.

---

### Official Review · Reviewer_SLRN · 2024-10-21

**Soundness:** 2
**Presentation:** 2
**Contribution:** 2
**Rating:** 6
**Confidence:** 3

**Summary:**

The paper introduces a new framework for estimating covariance in diffusion models, addressing issues in current methods such as high test-time computation and heavy approximations. The proposed method utilizes readily available covariance information from training data and the curvature of the generative trajectory. The authors also present a method for transferring covariance estimates across noise levels.

**Strengths:**

1. It is novel and reasonable to design low-dim expression and update rules for covariance approximation.
2. There are abundant math derivations to support the methods

**Weaknesses:**

1.  To handle linear inverse problems using diffusion models, there are a lot of classes of methods [1]. The experiments in this work only compare some of them, arthors should at least mension why take them out of comparison.
2. There lack of a experiment to back up whether the low-dim covariance of the proposed method can indeed match the underlying true one. I understand this cannot be done under high dimensional case, mayba a 2D or 10D suffice.
3. There is other applications of diffusion models would need covariance like causual reasoning [2], likelihood-evaluation [3][4] and adjoint guidance[5], maybe you could add a discussion to these.


[1] Daras, Giannis, et al. "A survey on diffusion models for inverse problems." arXiv preprint arXiv:2410.00083 (2024).

[2] Sanchez P, Liu X, O'Neil A Q, et al. Diffusion models for causal discovery via topological ordering[J]. arXiv preprint arXiv:2210.06201, 2022.

[3] Lu C, Zheng K, Bao F, et al. Maximum likelihood training for score-based diffusion odes by high order denoising score matching[C]//International Conference on Machine Learning. PMLR, 2022: 14429-14460.

[4] Anonymous. Gradient-Free Analytical Fisher Information of Diffused Distributions.

[5] Song K, Lai H. Fisher Information Improved Training-Free Conditional Diffusion Model[J]. arXiv preprint arXiv:2404.18252, 2024.

**Questions:**

1. refer to weaknesses.
2. in Figure 6, these setting of y is what? inpating transform or deblurring? Does this guidance meaningful across different y|x_0 and dataset?

---

> ### Author Response · Authors · 2024-11-20
> **Answer to reviewer SLRN**
>
> We thank the reviewer for their constructive feedback. We address the points that were raised below.
>
> **Choice of comparison methods.** Our focus on reconstruction guidance-based methods was motivated by several factors:
> 1. Direct evaluation of covariance estimation: These methods differ primarily in their covariance approximation strategies, allowing us to isolate the impact of our contribution
> 2. Theoretical connection: Our method builds on and extends the reconstruction guidance framework
> 3. Broad applicability: The insights from our covariance estimation technique could potentially benefit other method classes as well, and direct comparisons to them might not be as insightful.
>
> To provide broader context, we have now added comprehensive comparisons with DDNM+ [1] and DiffPIR [2] in Appendix H. We include results with extended step counts: 15, 30, 50, and 100 steps with both Heun and Euler solvers. Breakdown of the new results:
> - Our method outperforms the others with low step counts, and remains the best or second-best in LPIPS scores even at 100 steps, and competitive for the PSNR and SSIM metrics.
> - Note that DDNM+ is designed to enforce consistency with the measurement in cases where the measurement operator has a clearly defined nullspace, potentially driving the good performance in inpainting and super-resolution. In contrast, the reconstruction guidance methods perform consistently better on deblurring.
>
> **Lack of experiment on matching true covariance.** We have added a detailed quantitative analysis of the error of different covariance approximations for a 2D example in Appendix F. A highlight is that with a stochastic sampler, the accuracy improves with larger step counts. This is because low-rank updates are done in all directions due to noise in the sample path. In contrast, a deterministic solver has a minimum level of error it can reach.
>
> **Other applications of covariances.** We thank the reviewer for bringing these very relevant application areas to our attention. We have now added discussion to them in the Related Works section.
>
> **In Figure 6, the setting of y is what? Inpainting transform or deblurring?** The task we tested on was Gaussian blurring. We optimized the hyperparameters of the methods with hyperparameters on Gaussian blurring. We have updated the paper to explain this.
>
> We hope that this answers the main concerns, and we are open to further discussion.
>
> References:
> [1] Wang et al. "Zero-Shot Image Restoration Using Denoising Diffusion Null-Space Model."
>
> [2] Zhu et al. "Denoising Diffusion Models for Plug-and-Play Image Restoration."

---

> > ### Comment · Reviewer_SLRN · 2024-11-25
> >
> > Thanks for the clarification and extra experiments!
> >
> > The L-BFGS practice is reasonable. However, I am not confident about the linear inverse experiments part; I have very limited knowledge of this field.
> >
> > I will increase my score to 6.

---

### Official Review · Reviewer_nmFX · 2024-11-03

**Soundness:** 3
**Presentation:** 2
**Contribution:** 3
**Rating:** 6
**Confidence:** 4

**Summary:**

This paper introduces a new method to estimate the covariance during the denoising process in diffusion models. The key idea is using Tweedie's formula to relate the covariance of the denoiser with the Hessian of $\log(p(x_t))$, then using the gradient of $\log(p(x_t))$ (i.e. the score function) evaluated during each denoising step to estimate this Hessian. Starting with an estimate of the covariance of the data distribution, the algorithm performs low-rank updates to this covariance matrix, inspired by quasi-Newton methods, using the score function estimates given by the neural network. This method is then applied to fast sampling for solving inverse problems, showing improvements over other methods.

**Strengths:**

1. It is difficult to efficiently estimate the covariance of the denoiser due to its high-dimensionality, and methods that attempt to estimate it has have to use reduced-complexity representations. Using the DCT-diagonal basis to estimate the initial data covariance and a low-rank update inspired by quasi-Newton methods is a novel approach.

2. Because this method does not require training an additional neural network, it can be applied to existing pre-trained diffusion models to be used for solving inverse problems.

3. The experimental results seem promising and improves on previous methods on both qualitative examples and quantitative metrics.

**Weaknesses:**

1. There needs to be more theoretical or empirical justification for some of the algorithmic choices made, such as the choice of quasi-Newton method or the DCT basis used to approximate the data covariance.

2. The notation in the paper is inconsistent and the definition of some quantities are not clear, for example are $p(x, t)$, $p(x_t)$, $p_t(x)$ referring to the same quantity? Also, it is not clear what exactly are the full algorithms used for solving linear inverse problems; Algorithms 1 and 2 in the paper only updates the covariance estimate.

3. The experimental improvements for solving linear inverse problems seems to be only for the few-sample-step regime. It is unclear if this covariance estimation method can be helpful in other applications such as unconditional sampling, or with non-linear guidance terms.

**Questions:**

1. Since the data covariance is estimated in the DCT basis where the covariance is approximately diagonal, have you considered training a model to directly estimate the denoiser covariance in this basis, or with a diagonal + low-rank representation?

2. The numerical results in Table 1 seems to be worse than that in the DDNM paper, is this because the number of sampling steps is limited?

3. What is the computational overhead for estimating the data covariance and doing the low-rank updates, compared to the cost of evaluating the neural network?

4. What is the variance exploding case first mentioned in Line 213? This doesn't seem to be explicitly defined in the paper.

5. What is the difference between $\mu_{0\mid t}(x)$ and $\mu_{0 \mid t}(x_t)$, as well as $\sigma(t)$ and $\sigma_t$?

---

> ### Author Response · Authors · 2024-11-20
> **Answer to reviewer nmFX**
>
> We thank the reviewer for the positive comments and constructive feedback. They have prompted us to improve the writing provide more empirical evidence for the method. Below are our answers to the concerns and questions.
>
> **Theoretical justification for algorithmic choices:**
> There are specific reasons for our choices, and we have now elaborated more on them in Appendix H, with brief summaries in the main text. The reasons for our choices are:
> - **The DCT basis** has a firm grounding in the image processing literature. In the paper, we justified it with some empirical and theoretical results showing that image covariances are approximately diagonal in frequency bases [1]. It has also advantages over the DFT basis: It is purely real-valued, does not assume periodic boundaries, and in practice needs less coefficients to efficiently represent natural images (which is one of the reasons for its use in the JPEG compression standard).
> - **The BFGS update** has the attractive property of preserving positive-semidefiniteness (as opposed to, e.g., the symmetric rank-1 update). This combines well with performing the updates in the denoiser covariance, which is positive-definite (as opposed to the Hessian). Compared to Davidon-Fletcher-Powell (DFP), the difference is that the BFGS update minimizes a weighted Frobenius norm for the size of the update in inverse covariance [2], instead of the covariance directly. In the update formula in Eq.24., if we use A=I and the obseration noise is low, the inverse term is simply the inverse covariance. Thus, it could stabilise the updates across iterations, but this is more speculative, and DFP could work in practice as well.
>
> **Inconsistencies in notation and writing.** We went through the paper to unify the notation. Indeed, $p(x,t)$, $p(x_t)$, and $p_t(x_t)$ referred to the same thing. We replaced all uses of $p_t(x_t)$ with $p(x_t)$. We still use the $p(x_t)$ vs. $p(x,t)$ notation in the particular case of deriving the time update. Here, it is important to separate out the functional dependence on $t$ and $x$. For the rest of the paper, separating out $x_0$ and $x_t$ is needed instead. We now state this clearly in the text. Having $\mu_{0\mid t}(x_t), \mu_{0\mid t}(x)$, and $\sigma(t), \sigma_t$ were confusing as well, we now use $\mu_{0\mid t}(x_t)$ and $\sigma(t)$ consistently.
>
> **Additional experiments.** We have now conducted the following additional experiments, extending to the many-step regime (although still in the broad setting of linear inverse problems).
> - Results with the **Heun solver** for an extended number of steps: 15, 30, 50, and 100. Our method outperforms the others with low step counts, and remains the best or second-best in LPIPS scores even at 100 steps, and competitive for the PSNR and SSIM metrics.
> - Results with the **Euler solver** for 15, 30, 50, and 100 steps. The same conclusion holds here as well.
> - Both result tables include **two new models for comparison**: DDNM+ [4] (noise-resilient variant of DDNM) and DiffPIR [5].
>     - Especially on the Gaussian deblurring task, our method clearly outperforms DDNM+. In the low-step regime, it outperforms it across almost all tasks, and in the high-step regime, our model still outperforms the others in terms of LPIPS while remaining competitive in the other metrics.
>     - Note that DDNM+ is designed to enforce consistency with the measurement in cases where the measurement operator has a clearly defined nullspace. In contrast, the reconstruction guidance-based approaches do not explicitly consider this, explaining the performance gap in inpainting and super-resolution.
> - In response to reviewers SLRN and d5jC, we also added a quantitative analysis of the error of different covariance approximations in Appendix F.
> -  In principle, the method could be used for nonlinear inverse problems in the style of [3]. This would involve developing a method to sample from our low-rank Gaussian posterior approximation for estimating the integral in Eq.23. We consider this a promising direction for future work.
>
> **Have you considered training a model to directly estimate the denoiser covariance?** The paper focuses on training-free methods, which was motivated by two main considerations:
> 1) **Architectural.** Current neural net architectures do not process data in frequency spaces, and as such it is unclear what would be the best way to make them output frequency-diagonal covariances. In contrast, our method is applicable to any basis in a straightforward manner.
> 2) **Ease of use.** Our method is applicable to all standard diffusion models, without need for additional training.
>
> The work of [6] does consider trainable reverse covariances, and trains a model using a wavelet basis for the FFHQ dataset.
>
> **Comparison to DDNM.** See our earlier answer in "Additional experiments".

---

> > ### Author Response · Authors · 2024-11-20
> > **Continuing answer to reviewer nmFX**
> >
> > **Computational overhead.** The cost of data covariance estimation is negligible: We did it by performing the DCT transform for 1000 validation set images, and calculating the variances for each frequency in this basis, which can be done very quickly even on a CPU.
> >
> > Breaking down the computational cost of the method during sampling on an NVIDIA RTX4090, and pointing out easy further optimisations:
> >
> > - $\Pi$GDM method (15-step Heun): ~4.0s/sample
> > - Our DCT-Online method (15-step Heun): ~6.2s/sample
> >
> > The additional 2.2s overhead:
> > 1. Matrix inverses for guidance (Eq. 23): 1.5s
> >    - Can be optimized further via solver tolerance adjustment or through more efficient matrix inverse methods
> > 2. Covariance estimation with time and space updates: 0.7s
> >    - This cost is nearly eliminated if not using the space updates, since the matrix inverses required for the time updates become trivial with diagonal matrices.
> >    - These could also be parallelized with the guidance computation, since our current implementation does these on CPU. Moving to the GPU could also speed up computation.
> >
> > **Full algorithm for inverse problem.** We agree that this would be a significant improvement on the paepr. We have now added the full version of the algorithm in Appendix D. It contains both a basic implementation with an Euler solver, as well as an algorithm to apply it with any solver.
> >
> > **"Variance-exploding case" not defined.** Thank you for pointing this out. We meant the forward process induced by Eq. (1), where the variance of the noisy data increases with increasing diffusion time. We have now fixed this in the text.
> >
> > **References**
> > [1] Hyvärinen et al., "Natural Image Statistics: A Probabilistic Approach to Early Computational Vision."
> >
> > [2] Donald Goldfarb, "A Family of Variable-Metric Methods Derived by Variational Means"
> >
> > [3] Song et al., "Loss-Guided Diffusion Models for Plug-and-Play Controllable Generation."
> >
> > [4] Wang et al. "Zero-Shot Image Restoration Using Denoising Diffusion Null-Space Model"
> >
> > [5] Zhu et al. "Denoising Diffusion Models for Plug-and-Play Image Restoration."
> >
> > [6] Peng et al. "Improving Diffusion Models for Inverse Problems Using Optimal Posterior Covariance"

---

> > > ### Comment · Reviewer_nmFX · 2024-11-25
> > >
> > > Thank you for the detailed responses and making the changes to the paper. I will keep my score.

---

### Official Review · Reviewer_D6UH · 2024-11-04

**Soundness:** 4
**Presentation:** 2
**Contribution:** 3
**Rating:** 6
**Confidence:** 3

**Summary:**

The authors proposed a new methodological framework for estimating covariance in the generative methods of diffusion models.  The estimation of the covariance in the reverse diffusion process is purely based on the existing samples through a two-step updating scheme. The examples show the proposed method outperforms other existing methods in denoising.

**Strengths:**

Originality and significance: The proposed method is relevant to the diffusion models and is novel in that it provides a fast and efficient algorithm for estimating a broader class of covariance structures.

Quality and clarity: The paper clearly connects the proposed method to existing literature on similar problems.

**Weaknesses:**

The exposition can be improved as the current manuscript contains many inconsistent notations, undefined variables, and many typos. I would say it is a lovely work with appealing results, but unfortunately, the manuscript should be significantly revised and proofread to be easier to follow.

**Questions:**

1. In the image data results, adding the online update steps to either identity covariance or the FH covariance does not always improve the performance. Is there any explanation or further investigation on why it could happen?
2. Table 1 shows that in most cases, FH+online outperforms FH. But in Figure 7, only the denoised image from FH is demonstrated. Is there any vision difference between FH and FH+online in terms of the images in Figure 7?
3. Notational /exposition issues:
    1. In Eq. (1), the "noising forward process" should be defined in the joint distribution of the stochastic paths or a stochastic differential equation. Eq. (1) gives the margin distribution of the state at time t, which cannot formally "define" a process.
    2. In Eqs. (2) and (3), $\dot\sigma$ and $\omega_t$ are undefined.
    3. Above Eq. (7), $p_{0|t}$ is undefined.
    4. In Eq. (11), add a bracket to make the inverse clear. I.e., $[\nabla_{\mathbf{x}}^2\log p(\mathbf{x},t)]^{-1}$.
    5. The last few sentences in Sec. 3.1. confused me. The integral $\int p(\mathbf{x}')\mathcal N(\mathbf{x}'\mid \mathbf{0}, \sigma^2\mathbf{I})d\mathbf{x}'$ is not a convolution.
    6. In Eq. (18), should the right-hand side be $\left[\mathbf{\Sigma}_{0\mid t}(\mathbf{x}+\Delta\mathbf{x})\right]\Delta\mathbf{x}$.
    7. In Eq. (23), use either $\nabla_{\mathbf{x}}$ or $\nabla_{\mathbf{x_t}}$ consistently throughout the paper.
    8. In Eqs. (23) and (24), $\mathbf{A}$ is undefined.
    9. Above Eqs. (25), $\mu_{0\mid t}$ should be in boldface. (and many other occurrences throughout the paper).
    10. In Sec. 4.2., "we noticed that the guidance scale is overestimated the larger the dimensionality is, leading to overconfidence." does not read well --- the authors probably left over some words from a previous edition.
    11. Throughout the paper and the appendix, please only label the equations you refer to later.

---

> ### Author Response · Authors · 2024-11-20
> **Answer to reviewer D6UH**
>
> We thank the reviewer for their positive comments and the constructive suggestions to improve the writing. We address the questions and comments below.
>
> **Inconsistent notation, typos, variable definitions.** We went through the manuscript to address all the concerns raised, and make notation as consistent as possible. We have highlighted the more substantial changes in the text in green. We made a few exceptions:
> - Although we have done significant cleaning up, we still use the $p(x,t)$ notation when deriving the time update, because it is important to separate out the functional dependence on $t$ and $x$. For the rest of the paper, we use $p(x_t)$ since explictly defining the random variables $x_0$ and $x_t$ is needed for defining the posterior $p(x_0\mid x_t)$. We now added a paragraph about this in the beginning of the methods section. We are open to suggestions on alternative notation.
> - While we understand the point of minimising numbered equations, we believe that erasing the numbering in all unreferenced equations can be problematic as well. In particular, it may make it more difficult for future work to reference particular parts of the paper. As such, we only removed the labels for equations that clearly serve no other purpose than to lead to another equation, or labeled lines that were not proper equations (an equation split into two lines). We are open to more discussion.
>
> **Whether online updates always improve performance.** There are a few possible reasons for why they do not always result in improved performance:
> - The approximate nature of the Gaussian approximation means that we do not have a strict guarantee that a more accurate covariance always leads to better results in PSNR, SSIM or LPIPS scores. While our conclusion is that large improvements in the accuracy clearly improve performance (comparing the identity base covariance to DCT base covariance), it may also be the case that more modest improvements in covariance estimation only improve results on average across tasks, in contrast to improving every instance.
> - The method itself is approximate, and inaccuracies in the finite-difference updates to the covariance could sometimes cause inaccuracies in the covariance estimate.
>
> **FH+Online result visualisation.** We originally left FH+Online out of the image due to space reasons, but agree that it helps to have it as well. The results are not visually significantly different, although FH+Online does perform better on average. For the 30-step case, the generated images have slightly less details than with FH, possibly explaining the good PSNR and SSIM performance, but slightly worse LPIPS performance of FH+Online.

---

> > ### Comment · Reviewer_D6UH · 2024-11-25
> >
> > Thank you for the comments. I will keep my scores.

---

### Official Review · Reviewer_ei58 · 2024-11-05

**Soundness:** 3
**Presentation:** 2
**Contribution:** 2
**Rating:** 6
**Confidence:** 2

**Summary:**

The authors introduce a new covariance estimation method which makes use of covariance information in denoiser and the trajectory curvature without introducing significant additional compute. Moreover, to make the approach suitable for high-dimensional data (e.g. covariance matrices storage issue), the low-rank updates is proposed.  The authors validate their approach on linear inverse problems, demonstrating its effectiveness compared to baselines under four metrics.

**Strengths:**

- Novel method, proposed a covariance estimate method via separately updating along time and position/space. Also provide a practical implementation for the proposed method.
- The paper provides theoretical insights into why accurate covariance estimation is crucial for unbiased conditional generation, which is supported by experimental results on inverse problems.

**Weaknesses:**

- The experiment section is not as strong as methodology section: only one dataset and one ODE denoiser is tested under low NFE setting. The design of the experiment and the corresponding results analysis can be improved.
- Only one ODE sampler (Heun) with steps=15, 30 is tested. While indeed results show the importance of accurate covariance estimate, it’s hard to conclude that this is applicable to all standard diffusion models.

**Questions:**

- The authors assume that the observation model p(y|x0) is linear gaussian. Is there any other real world problems can be expressed in this form?
- Continue with weakness section: Can you provide more experimental results?
For example, does the performance changes when using more Heun steps? does the improvement also happen to other type of discretization methods, e.g. Euler? Also there are many other ODE samplers and SDE samplers, if you change the ODE samplers with SDE ones, does the introduced stochasticity make any difference when using your covariance estimate method?

---

> ### Author Response · Authors · 2024-11-20
> **Answer to reviewer ei58**
>
> We thank the reviewer for the positive feedback, but also for the suggestions regarding additional experiments. This motivated us to improve the experiments section. We respond to each of the raised points below.
>
> **New experiments.** We agreed that the experiments section would be stronger with a more thorough analysis. We have now added the following results in Appendix G:
> - Results with the Heun solver for an extended number of steps: 15, 30, 50, and 100. Our method generally outperforms the others with low step counts, and remains the best or second-best in LPIPS scores even at 100 steps, and competitive for the PSNR and SSIM metrics. This suggests tha that while the importance of accurate covariance estimates diminishes with increasing step counts, it does not disappear entirely.
> - Results with the Euler solver for 15, 30, 50, and 100 steps. The same conclusion holds here as well.
> - Both result tables include two new models: DDNM+ [1] and DiffPIR [2].
>     - Note that DDNM+ is designed to enforce consistency with the measurement in cases where the measurement operator has a clearly defined nullspace, and as such performs well in inpainting and super-resolution where this is the case. For deblurring, however, the other methods are more robust.
> - In response to reviewers SLRN and d5jC, we also added a quantitative analysis of the error of different covariance approximations in Appendix F. We also did analysis with stochastic solvers in this context, and we show that they have the advantage of converging to the correct covariance with large step counts. For the high-dimensional image experiments, our focus was on fast solvers instead, which is why we focused on deterministic methods that are better at low step counts [9].
>
> **Whether linear-Gaussian observation models are realistic.** The linear-Gaussian observation model, while seemingly restrictive, has wide applications that have been the subject of much study in inverse problems in different disciplines of science and engineering [3]. Examples include:
> - Medical imaging problems, like Computed Tomography [4] and MRI reconstruction [5]
> - The field of compressed sensing [6]
> - Seismic inversion [7], i.e., determining the physical properties under the surface of the Earth with indirect measurements
> - Radio interferometry [8]
> - As considered in this paper, image deblurring, inpainting, and super-resolution.
>
> Please, let us know if you have further concerns.
>
> References:
>
> [1] Wang et al. "Zero-Shot Image Restoration Using Denoising Diffusion Null-Space Model"
>
> [2] Zhu et al. "Denoising Diffusion Models for Plug-and-Play Image Restoration."
>
> [3] Tarantola, Albert. "Inverse problem theory and methods for model parameter estimation."
>
> [4] Hsieh, Jiang. "Computed tomography: principles, design, artifacts, and recent advances."
>
> [5] Liang, Zhi-Pei, and Paul C. Lauterbur. "Principles of magnetic resonance imaging."
>
> [6] Duarte, Marco F., and Yonina C. Eldar. "Structured compressed sensing: From theory to applications."
>
> [7] Cooke, D. A.; Schneider W. A. "Generalized linear inversion of reflection seismic data."
>
> [8] Wiaux et al. "Compressed sensing imaging techniques for radio interferometry."
>
> [9] Karras et al., "Elucidating the Design Space of Diffusion-Based Generative Models"

---

> > ### Author Response · Authors · 2024-11-27
> >
> > Dear reviewer ei58,
> >
> > We would like to thank again for your time in reviewing our paper, and providing constructive criticism that has helped to improve our paper. Would you have the time to indicate whether we have addressed your concerns (to raise the score)? We are also happy to discuss more.

---

> > > ### Author Response · Authors · 2024-11-28
> > > **Additional results on a different denoiser network**
> > >
> > > To provide further experimental evidence, we now managed to test the model with another denoiser network, trained on the FFHQ dataset, and obtained from [1]. We performed a similar sweep as for the ImageNet model with 15, 30, 50 and 100 steps, with the Euler sampler (we chose Euler due to faster generation time here). The results table is appended in Appendix H. The conclusions are largely the same as for ImageNet: On low step counts, our models are generally the best, while the improvement becomes smaller on large step counts especially in the PSNR and SSIM metrics. Our method remains competitive to the best models, however, even on large step counts.
> > >
> > > We think that this is a useful addition to the paper, to showcase that the method works with different types of data sets and different denoisers. We would again like to thank the reviewer for pushing us to improve the experimental consolidation of the method. We believe that this has significantly strengthened the message of the paper.
> > >
> > > **References**
> > >
> > > [1] Chung et al., "Diffusion Posterior Sampling for General Noisy Inverse Problems", ICLR 2023

---

> > > > ### Comment · Reviewer_ei58 · 2024-11-28
> > > > **Thanks to the response**
> > > >
> > > > Thank you for the additional results. I think my concerns are resolved and I am happy to increase my score.
> > > > Good luck!

---

### Author Response · Authors · 2024-11-25
**Response to all reviewers**

We would like to thank everyone for their efforts in reviewing our paper and all the constructive suggestions to improve it. As the discussion period is soon finishing and the paper is borderline with the current scores, **we would appreciate if you have the time to look at the responses and indicate whether they answered the concerns raised.**

We also gather together common concerns and the corresponding improvements to the paper we have done during the rebuttal.

**Writing improvements.** The most common concern was that the writing could be improved (D6UH, nmFX, d5jC). In particular, issues with notation inconsistency (D6UH,nmFX) were raised, and the ambiguity of the phrase 'conditional generation' (d5jC). We have now fixed the inconcistencies (with any changes in notation explicitly motivated in the text), and resolved the ambiguity in the beginning of the text. We have also fixed various smaller issues raised by the reviewers.

**Wider experiments on inverse problems.** A common (ei58, nmFX) concern was the limited number of high-dimensional experiments. In response to this, we have significantly increased the amount of results in the paper in **Appendix G**:
- **Results on 50 and 100 steps.** We added results with higher step counts. While our lead on PSNR and SSIM does not hold for these higher step counts, our results are usually still the best in the LPIPS metric, which is known to correspond better to human vision than PSNR and SSIM [1].
- **Results with the Euler solver.** We also added all results with the Euler solver, with the same conclusions.
- **New comparison models.** We added influential recent works DDNM [2] and DiffPIR [3] to all our tables. The overall conclusion that our model is the best for low step counts remains. For large step counts (50, 100), it is still usually the best in the LPIPS score. Note that we previously focused on reconstruction guidance methods to enable direct reasoning on the effect of the covariance estimate.

**Theoretical analysis / direct error analysis of the covariance estimates** Reviewer d5jC pointed out the lack of theoretical analysis on the error caused by the method. Reviewer SLRN similarly pointed out that analysis on the error of the covariance estimates would be beneficial for the paper. Following this, we have added an empirical analysis in **Appendix H** that show our method
- **converges to the true covariance** with increasing step counts for a stochastic solver.
- **improves the covariance estimate accuracy** compared to simpler heuristics
- **improves over just initializing the denoiser covariance at data covariance** when applying the space updates

 **Full algorithm.** We have added full pseudo-code algorithms for implementing the method with reconstruction guidance with Gaussian observations in **Appendix D**. This was raised by reviewer nmFX.

**References**

[1] Zhang et al., "The Unreasonable Effectiveness of Deep Features as a Perceptual Metric"

[2] Wang et al. "Zero-Shot Image Restoration Using Denoising Diffusion Null-Space Model"

[3] Zhu et al. "Denoising Diffusion Models for Plug-and-Play Image Restoration."

---

### Meta-Review · Area_Chair_eDKF · 2024-12-21

**Metareview:**

The paper focuses on the importance of estimating covariance for clean data from noisy observations in training-free guided generation methods for diffusion models. It highlights the limitations of existing approaches, which involve significant test-time computation, modifications to the standard diffusion training process or denoiser architecture, or reliance on heavy approximations. To address these challenges, the authors propose a novel framework leveraging covariance information readily available from training data and the curvature of the generative trajectory, linked via the second-order Tweedie's formula. Their method integrates this information through (i) a new technique for transferring covariance estimates across noise levels and (ii) low-rank updates within a specific noise level. Validation on linear inverse problems demonstrates superior performance over recent baselines, particularly with fewer diffusion steps.

Based on the reviews the strengths and weakness of the paper are as follows:

Pros:

+Introduces an efficient covariance estimation method using Tweedie’s formula and low-rank updates without additional training.

+nice mathematical foundations connecting curvature and covariance.

+Outperforms baselines in linear inverse problems, particularly with fewer steps.

+Works with pre-trained diffusion models, showcasing low computational overhead.

+Addressed reviewer feedback on clarity, notation, and added broader experiments.

Cons:

-Scope Limitations: Focuses only on linear inverse problems; broader applicability remains untested.

-Presentation Issues: Initial manuscript had inconsistent notation and unclear definitions.

-Lack of Theoretical Bounds: No analysis of approximation errors in the method.

-Limited Comparisons: Initially lacked comprehensive baselines, later added during rebuttal.

The authors actively addressed most reviewer concerns through additional experiments, theoretical analysis, and manuscript revisions. These efforts improved the paper's clarity and strengthened its empirical and theoretical contributions. As a result some reviewers raised their score and now all reviewers recommend acceptance. I concur.

**Additional Comments On Reviewer Discussion:**

The authors actively addressed most reviewer concerns through additional experiments, theoretical analysis, and manuscript revisions. These efforts improved the paper's clarity and strengthened its empirical and theoretical contributions. As a result some reviewers raised their score and now all reviewers recommend acceptance. I concur.

---

### Decision · Program_Chairs · 2025-01-22

Accept (Poster)